# INTERLCM: LOW-QUALITY IMAGES AS INTERMEDIATE STATES OF LATENT CONSISTENCY MODELS FOR EFFECTIVE BLIND FACE RESTORATION

**Senmao Li**[1,2*] **Kai Wang**[2†] **Joost van de Weijer**[2] **Fahad Shahbaz Khan**[3,4]
**Chun-Le Guo**[1,6] **Shiqi Yang**[5] **Yaxing Wang**[1,6] **Jian Yang**[1] **Ming-Ming Cheng**[1,6]
[1]VCIP, CS, Nankai University  [2]Computer Vision Center, Universitat Autònoma de Barcelona
[3]Mohamed bin Zayed University of AI  [4]Linkoping University  [5]SB Intuitions, SoftBank
[6]Nankai International Advanced Research Institute (Shenzhen Futian), Nankai University
{senmaonk, shiqi.yang147.jp}@gmail.com, {kwang, joost}@cvc.uab.es
fahad.khan@liu.se, {guochunle, yaxing, csjyang, cmm}@nankai.edu.cn

## ABSTRACT

Diffusion priors have been used for blind face restoration (BFR) by fine-tuning diffusion models (DMs) on restoration datasets to recover low-quality images. However, the naive application of DMs presents several key limitations. (i) The diffusion prior has inferior semantic consistency (e.g., ID, structure and color.), increasing the difficulty of optimizing the BFR model; (ii) reliance on hundreds of denoising iterations, preventing the effective cooperation with perceptual losses, which is crucial for faithful restoration. Observing that the latent consistency model (LCM) learns consistency noise-to-data mappings on the ODE-trajectory and therefore shows more semantic consistency in the subject identity, structural information and color preservation, we propose *InterLCM* to leverage the LCM for its superior semantic consistency and efficiency to counter the above issues. Treating low-quality images as the intermediate state of LCM, *InterLCM* achieves a balance between fidelity and quality by starting from earlier LCM steps. LCM also allows the integration of perceptual loss during training, leading to improved restoration quality, particularly in real-world scenarios. To mitigate structural and semantic uncertainties, *InterLCM* incorporates a Visual Module to extract visual features and a Spatial Encoder to capture spatial details, enhancing the fidelity of restored images. Extensive experiments demonstrate that *InterLCM* outperforms existing approaches in both synthetic and real-world datasets while also achieving faster inference speed. Project page: https://sen-mao.github.io/InterLCM-Page/

## 1 INTRODUCTION

Blind face restoration (BFR) aims to restore high-quality (HQ) images from low-quality (LQ) input that exhibit complex and unknown degradation, such as down-sampling (Chen et al., 2018; Bulat et al., 2018), blurriness (Zhang et al., 2017; 2020; Wang et al., 2024a), noise (Dogan et al., 2019), compression (Dong et al., 2015), etc. BFR has undergone significant advances in recent years. Existing methods primarily focus on learning a direct mapping between LQ and HQ images, often incorporating various priors to enhance restoration performance. Early works mainly explore geometric priors, such as facial landmarks (Chen et al., 2018), parsing maps (Chen et al., 2021; Shen et al., 2018), and heat maps (Yu et al., 2018), to offer explicit information about face restorations. Reference prior (Gu et al., 2022; Zhou et al., 2022) methods are taking additional high-quality images to enhance the restoration of LQ images. More recently, generative priors (Wang et al., 2021a; Yang et al., 2021) have been widely used in blind face restoration to obtain realistic textures.

With the superior generative capabilities of recent successful diffusion models (Ramesh et al., 2022), which are trained on billions of data (Schuhmann et al., 2022), the diffusion-prior methods (Wang

---

*Work done during a research stay at Computer Vision Center, Universitat Autònoma de Barcelona.
†The corresponding author.

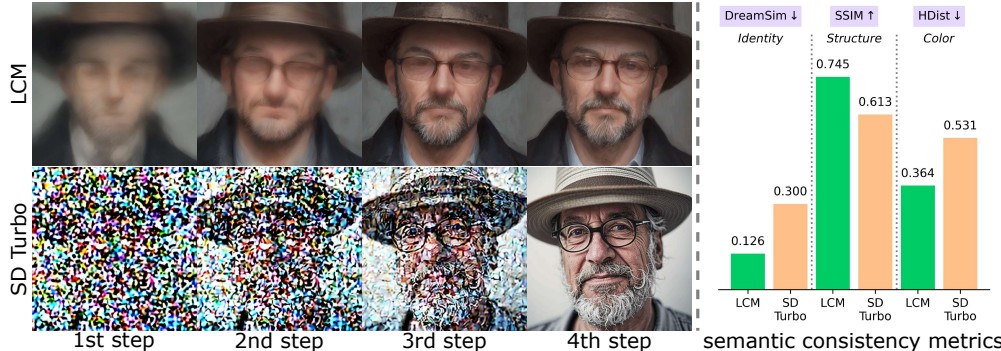

Figure 1: (*Left*) The intermediate states in 4-step LCM and SD Turbo models. The network used in LCM maps to the real image space, while SD Turbo progressively denoises the noisy image. (*Right*) Given the prompt "A headshot of a man with hat and glasses", we generate 1000 images with both LCM and SD Turbo models. Then we use DreamSim, SSIM, and color histogram distance (HDist) to measure the semantic consistency in the subject identity, spatial structure and color preservation.

et al., 2023; Miao et al., 2024; Lu et al., 2024) have been explored to solve the BFR problem. Although reasonable restoration results are achieved, existing diffusion-based methods (Wang et al., 2021a; Yue & Loy, 2024) generally suffer from several major limitations. (i) The diffusion prior has inferior semantic consistency, namely identity consistency, structural stability, color preservation, etc. which increases the difficulty of optimizing the BFR model(Zhou et al., 2022). As an example, we evaluate the semantic consistency between the estimated real image in each step for a conventional diffusion model SD Turbo (Sauer et al., 2023)[1] and the latent consistency model (LCM) (Luo et al., 2023a), as shown in Fig. 1. It is evident that the conventional diffusion models exhibit weaker semantic consistency prior information compared with the consistency models. (ii) Diffusion-based methods that rely on standard diffusion models face challenges in sampling, as they require many iterations to produce the real image outputs. They cannot easily incorporate with a perceptual loss applied to the final image outputs. Despite existing methods (Chung et al., 2023; Laroche et al., 2024) compute the perceptual loss with real images obtained from the intermediate step, these real images show a appearance gap compared to the final image output (see Appendix E.6 for details).

To address these problems, we introduce the latent consistency model (LCM) into blind face restoration tasks, which has not been explored before. More specifically, the LCM model learns to map any point on the ODE (Song et al., 2023) trajectory to its origin for generative modeling. That property differs significantly from the conventional diffusion models, where the iterative sampling process progressively removes noise from the random initial vectors. Based on the LCM property, we propose our method *InterLCM*, which regards the LQ image as the input in an *intermediate step of LCM models* and obtains the high-quality image by performing the remaining few denoising steps (i.e., 3 steps) in 4-step LCM. By this means, *InterLCM* maintains better semantic consistency originated from the LCM. Meanwhile, benefitting from this property, we can integrate with both perceptual loss (Johnson et al., 2016) and adversarial loss (Goodfellow et al., 2014), which are commonly used in restoration model training, leading to a high-quality and high-fidelity face restoration output.

However, directly applying the LCM to blind face restoration brings randomness to the generated structures and semantics, which originate from the random sampling paths (see Sec. 3.2 and Fig. 5). We therefore propose to apply two extra components to *InterLCM*. First, a CLIP image encoder and Visual Encoder as Visual Module that helps to extract semantic information from faces, providing the LCM with face-specific priors. Second, to prevent changes in content (e.g., structure), we include a Spatial Encoder to leverage the strong semantic consistency of the LCM model. More specifically, we follow the ControlNet architecture design to copy the UNet encoder part as the Spatial Encoder. Note that the Spatial Encoder differs from the ControlNet by the training schemes, where it is commonly trained with the diffusion loss while our Spatial Encoder backpropagates from the real image (through the denoising steps) to the initial low-quality image. During this process, the Visual Encoder and Spatial Encoder are updated with gradients.

---

[1]We regard SD Turbo as a typical representative of the diffusion models, since it inherits the characteristics of the diffusion model well. While LCM is distilled with the consistency regularization.

In the experiments, we performed extensive experiments to compare *InterLCM* with existing approaches, on synthetic and real-world datasets including CelebA, LFW, WebPhoto, etc. Our method achieves better qualitative and quantitative performance while also achieving faster inference times. In summary, our work makes the following contributions:

- We introduce *InterLCM*, a simple but effective BFR framework leveraging the latent consistency model (LCM) priors. By considering the low-quality image as the intermediate state of LCM models, we can effectively maintain better semantic consistency in face restorations.

- Using LCM mapping each state to the original image level point, our method *InterLCM* has additional advantages: few-step sampling with much faster speed and integrating our framework with commonly used perceptual loss and adversarial loss in face restoration.

- Through extensive experiments over synthetic and real image datasets, we demonstrate the effectiveness and authenticity of our *InterLCM* in restoring HQ images, especially in real-world scenarios with unpredictable degradations.

## 2 RELATED WORK

### 2.1 BLIND FACE RESTORATION.

In real-world scenarios, face images may suffer from various types of degradation, such as noise, blur, down-sampling, JPEG compression artifacts, and etc. Blind face restoration (BFR) aims to restore high-quality face images from low-quality ones that suffer from unknown degradation. The BFR approaches are mainly focused on exploring better face priors, including geometric priors, reference priors, and generative priors. Diffusion prior, which is more explored in recent years, belongs to a broader stream of generative priors. For the geometric-prior methods, they explore the highly structured information in face images. The structural information, such as facial landmarks (Chen et al., 2018), face parsing maps (Shen et al., 2018; Chen et al., 2021; Su et al., 2023) and 3D shapes (Hu et al., 2020; Zhu et al., 2022; Lu et al., 2024), can be used as a guidance to facilitate the restoration. However, since the geometric face priors estimated from degraded inputs can be unreliable, they may lead to the suboptimal performance of the subsequent BFR task. Some existing methods (Dogan et al., 2019; Li et al., 2018) guide the restoration with an additional HQ reference image that owns the same identity as the degraded input, which is referred to as the reference-priro BFR approaches. The main limitations of these methods stem from their dependence on the HQ reference images, which are inaccessible in some scenarios. More recent approaches directly exploit the rich priors encapsulated in generative models for BFR, which are denoted as generative priors.

**GAN-prior.** By applying the GAN inversion (Xia et al., 2022), the earlier generative-prior explorations (Gu et al., 2020; Menon et al., 2020) iteratively optimize the latent code of a pretrained GAN for the desirable HQ target. To circumvent the time-consuming optimization, some studies (Yang et al., 2021; Chan et al., 2021) directly embed the decoder of the pre-trained StyleGAN (Gal et al., 2021) into the BFR network and evidently improve the restoration performance. The success of VQ-GAN (Crowson et al., 2022) in image generation also inspires several BFR methods to design various strategies (Wang et al., 2022; Zhou et al., 2022) to improve the matching between the codebook elements of the degraded input and the underlying HQ image.

**Diffusion-prior.** Recently, the diffusion model has been proven to be more stable than GAN (Dhariwal & Nichol, 2021), and the generating images are more diverse. This has also received attention in the blind face restoration task. IDM (Zhao et al., 2023) introduces an extrinsic pre-cleaning process to further improve the BFR performance on the basis of SR3 (Saharia et al., 2022). To accelerate the inference speed, LDM (Rombach et al., 2022) proposed to train the diffusion model in the latent space. In a bid to circumvent the laborious and time-consuming retraining process, several investigations (Lin et al., 2023; Wang et al., 2023) have explored the utilization of a pre-trained diffusion model as a generative prior to facilitate the restoration task. More specifically, DiffBIR (Lin et al., 2023) and SUPIR (Yu et al., 2024) leverage the pretrained Stable Diffusion (Rombach et al., 2022) as the generative prior, which can provide more prior knowledge than other existing methods. DR2 (Wang et al., 2023) and CCDF (Chung et al., 2022) diffuse input images to a noisy state where various types of degradation have weaker scales than the added Gaussian noises, following by capturing the semantic information during denoising steps. Moreover, this restoration using noisy states (Wang et al., 2023; Chung et al., 2022) or diffusion bridges (Liu et al., 2023) can accelerate the

inference. The common idea underlying these approaches is to modify the reverse sampling process of the pre-trained diffusion model by introducing a well-defined or manually assumed degradation model as an additional constraint. Even though these methods perform well in certain ideal scenarios, they can not deal with the BFR task since its degradation model is unknown and complicated.

However, these diffusion-prior based approaches still suffer from time-consuming inferences since the diffusion models have to pass through multiple steps. Furthermore, they mostly can only be trained with the reconstruction loss succeeded from the latent diffusion training. The common used perceptual loss in image restoration tasks cannot be well integrated in their frameworks, which may lead to suboptimal perceptual generation with these methods.

## 2.2 TEXT-TO-IMAGE GENERATIVE MODELS

Diffusion models (Shonenkov et al., 2023; Ho et al., 2022; Chen et al., 2023) have emerged as the new state-of-the-art models for text-to-image generation. They commonly involve encoding text prompts utilizing a pre-train language encoder, such as CLIP (Radford et al., 2021) and T5 (Raffel et al., 2020). The output is subsequently inserted into the diffusion model through the cross-attention mechanism. For base architectures, UNet (Ronneberger et al., 2015) and DiT (Peebles & Xie, 2023) are widely adopted. In this paper, we mainly build our method on the Stable Diffusion (Rombach et al., 2022) model as a powerful representative generative model of T2I generation models.

**Distillation of T2I models.** The diffusion models are bottlenecked by their slow generation speed. Recently, the distillation-based technique (Hinton et al., 2014) has been widely used in the acceleration of diffusion models. The student model distilled from a pretrained teacher (Luo et al., 2023a; Sauer et al., 2023) generally has faster inference speeds. Earlier studies (Salimans & Ho, 2022; Meng et al., 2023) utilize progressive distillation to gradually reduce the sampling steps of student diffusion models. Also, The sampling time of the pretrained teacher models are hampering training efficiency. To address this limitation, several works (Gu et al., 2023; Nguyen & Tran, 2023) propose using various bootstrapping techniques. For instance, Boot (Gu et al., 2023) is trained using bootstrapping based on two consecutive sampling steps, achieving image-free distillation. SDXL-Turbo (Sauer et al., 2023) introduces a discriminator and combines it with score distillation loss.

**Additional image control of T2I models.** Text descriptions guide the diffusion model in generating images but are insufficient in fine-grained control over the generated results. The fine-grained control signals are diverse in modality, including layouts, segmentations, depth maps, etc. Considering the powerful generation ability of the T2I model, there have been a variety of methods (Li et al., 2024a; Zavadski et al., 2023; Lin et al., 2024) dedicated to adding image controls to the T2I generative models. As a representative, ControlNet (Zhang et al., 2023) proposes using the trainable copy of the UNet encoder in the T2I diffusion model to encode additional condition signals into latent representations and then applying zero convolution to inject into the backbone of the UNet in diffusion modal. The simple but effective design shows generalized and stable performance in spatial control and thus is widely adopted in various downstream applications. Similarly, the T2I-Adapter (Mou et al., 2024) trains an additional controlling encoder that adds an intermediate representation to the intermediate feature maps of the pre-trained encoder of Stable Diffusion.

Nonetheless, the T2I models with additional image conditions are still generating images from Gaussian noises. How to explore their possibilities in solving image restoration tasks is still not explored. In this paper, we successfully make them start the generation from degraded low-quality images to restore the high-quality images and merged them together with the acceleration T2I models.

## 3 METHOD

BFR aims to restore a HQ image from its LQ counterpart while preserving semantic consistency with the LQ image under unknown and complex degradation. In this section, we first introduce the preliminaries on latent diffusion models and latent consistency models in Sec. 3.1. Then we detail our method, *InterLCM*, in Sec. 3.2. In *InterLCM*, following the LCM noise addition process, we begin by investigating the intermediate state of the LCM to which the LQ image should be regard as. We then introduce Visual Module and Spatial Encoder to preserve the semantic information and structure in the reconstructed HQ image. The illustration of *InterLCM* is shown in Fig. 3 and Algorithm 1 in Appendix B.

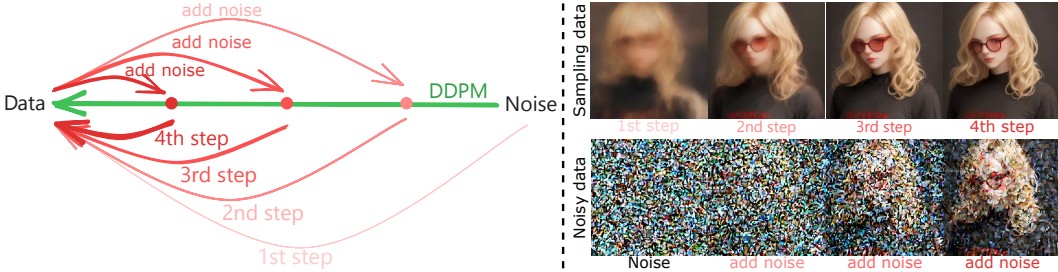

Figure 2: (*Left*) The 4-step LCM map its origin at each sampling step: Noise $\xrightarrow{\text{1st step}}$ Sampling data $\xrightarrow{\text{add noise}}$ Noisy data $\xrightarrow{\text{2nd step}}$ Sampling data $\xrightarrow{\text{add noise}}$ Noisy data $\xrightarrow{\text{3rd step}}$ Sampling data $\xrightarrow{\text{add noise}}$ Noisy data $\xrightarrow{\text{4th step}}$ Sampling data. In the first step, the origin image is predicted from random noise. In each remaining step, noise is added to the origin image produced in the previous step. (*Right*) The predicted origin images are shown for each step (the first row). The random noise and noisy data from the first to third steps (the second row). For example, given one prompt case "blond woman with red glasses and a black shirt", the generated image at each step shows semantic consistency in the subject identity, structural information and color constancy (the first row).

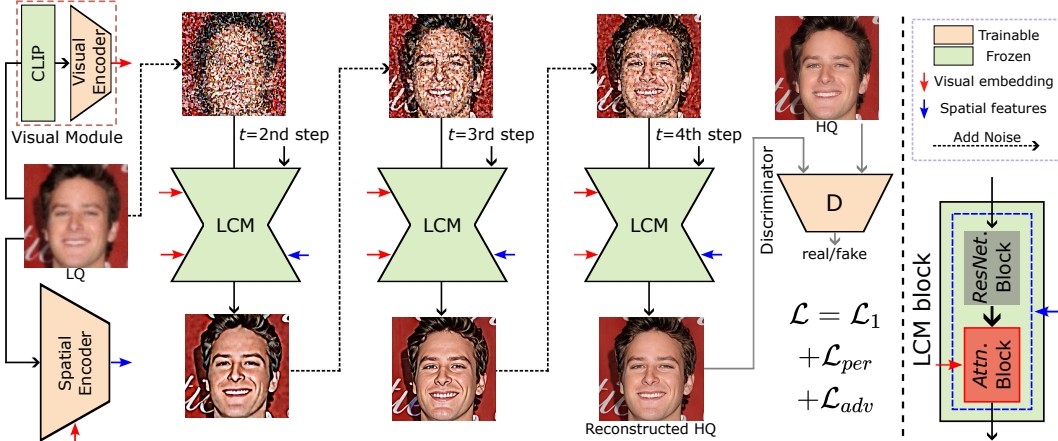

Figure 3: Overview of the proposed *InterLCM* framework. The Visual Module takes LQ images to output the visual embeddings. A Spatial Encoder is used to provide structure information. We consider the LQ image as the intermediate state of LCM. Through standard LCM conditioned with both the visual embedding and spatial features, the LQ input can be reconstructed as a HQ image.

## 3.1 PRELIMINARIES

**Latent Diffusion Models.** To enable diffusion model (DM) trained over limited computing resources while retaining the generation quality, Latent Diffusion Models (LDMs) (Rombach et al., 2022) encode an image $x$ into a latent representation $z_0$ using an encoder $\mathcal{E}$ and reconstruct it using a decoder $\mathcal{D}$. The LDMs aims to train a noise prediction network $\epsilon_\theta$ with diffusion loss:

$$\mathcal{L} = \mathbb{E}_{z_0, t, \epsilon \sim \mathcal{N}(\mathbf{0}, \mathbf{I})} \|\boldsymbol{\epsilon} - \boldsymbol{\epsilon}_\theta(z_t, c, t)\|_2^2 \tag{1}$$

In the diffusion inference phase, a LDM predicts noise using the pretrained denoising network $\boldsymbol{\epsilon}_{\boldsymbol{\theta}}(z_t, c, t)$ with the text condition $c$, resulting in a latent $z_{t-1}$ following the DDPM scheduler (Ho et al., 2020) (see Fig. 2 (left), the green arrow line). The final latent $z_0$ is obtained sequentially.

**Latent Consistency Models.** Consistency Models (CMs) (Song et al., 2023) adopt consistency mapping to directly map any point in ODE trajectory back to its origin, facilitating semantic consistency generation compared to LDMs. A LCM $\boldsymbol{f_\theta}(z_{\tau_n}, c, \tau_n)$ can be distilled from a pretrained LDM (e.g., Stable Diffusion (Rombach et al., 2022)) using the consistency distillation loss (Song et al., 2023) for few-step inference, where $c$ is the given text condition. LCM directly predicts the origin $z_0$ of augmented PF-ODE trajectory (Luo et al., 2023a), generating samples in a single step.

The LCM enhances sample quality while maintaining semantic consistency by alternating between denoising and noise addition steps (see Fig. 2 (left), the various red arrow lines). Specifically, in the $n$-th iteration, the LCM first applies a noise addition forward process to the previously predicted sample $z_0 = \boldsymbol{f_\theta}(z_{\tau_{n+1}}, c, \tau_{n+1})$, resulting in $z_{\tau_n}$. Here, $\tau_n$ represents a decreasing sequence of time steps, where $n \in \{1, \cdots, N-1\}$, $\tau_1 > \tau_2 > \cdots > \tau_{N-1}$, and $N$ ($N = 4$) is the number of steps in the LCM. Then, the prediction for the next $z_0 = \boldsymbol{f_\theta}(z_{\tau_n}, c, \tau_n)$ is carried out again.

## 3.2 *InterLCM*: Low-Quality images as Intermediate states of LCM

Our proposed *InterLCM* is built on the LCM model. As shown in Fig. 3, the random noise is added to LQ image $x_l$, which already contains complex and unknown degradation. The Visual Module takes LQ image as input and returns the visual embedding, which replaces the text embedding used in the standard LCM to supply the face-specific semantic information. To preserve the structure of LQ image, we utilize a Spatial Encoder to provide LCM with structure information. Through standard LCM processing with both visual embedding and spatial features, the LQ input can be reconstructed into an HQ output. In this subsection, following the LCM noise addition process, we begin by investigating which intermediate state of the LCM to insert LQ image. We then detail Visual Module and Spatial Encoder.

**2nd-step intermediate state.** To leverage the content consistency inherent in LCM (Luo et al., 2023a), we retain the pretrained model and follow its sampling process. As shown in Fig. 2 (right, the first row), the 4-step LCM sampling process generates semantic consistency images. In the first step, LCM directly predicts an image from random noise. In each remaining step, LCM first adds noise to the previous image and then predicts a finer output. Based on the three noise addition processes in each of the 4-step LCM, we first move the LQ image to each intermediate state of LCM. As shown in Fig. 4, we empirically find that the distribution of the LQ image is closer to that of the generated image after the first noise addition

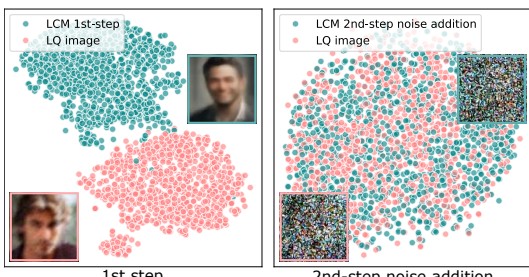

Figure 4: t-SNE visualizations of feature distributions show the first step sampling similarity of LCM and the LQ image (FID=103.70), and their noisy intermediate states after LCM 2nd-step noise diffusion (FID=2.83).

(second step noise addition) than other intermediate states (see Appendix C.1 for more detail). Therefore, we use the LQ image as the intermediate state after the first noise addition in LCM. Subsequently, the LCM is applied starting from *the second step*.

**Visual Encoder.** Ideally, the model should reconstruct image quality and align semantic information with the LQ image. However, noise diffusion introduces randomness, altering the original semantics of the LQ image, regardless of whether the prompt is a null-text or text prompt. For example, as shown in Fig. 5, when given a LQ image and a null-text prompt (i.e.,

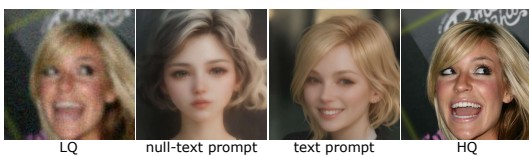

Figure 5: Naive LCM alters the original semantics of the LQ image (e.g., hair).

$\varnothing = $""), the hair color changes to white in the generated image (Fig. 5 (the second column)). Even given a text prompt (that is, "a woman with blonde hair and a smile" [2]) obtained from the HQ image, the straight hair changes to curly in the generated image (Fig. 5 (the third column)).

To provide LCM with face-specific prior to produce semantic consistent content, we propose to use a Visual Module (Fig. 3). The Visual Module provides face-specific semantic information to the pretrained LCM, similar to how text prompts are used in standard text conditioned image generation (Luo et al., 2023a). We employ visual embedding, first extracting general CLIP visual features (Radford et al., 2021) from LQ image $x_l$, which are then distilled by the Visual Encoder (VE) to yield face-specific semantic information, defined as $c_v = VE(CLIP(x_l))$. This approach aligns $c_v$ with the text embedding the LCM typically uses for its text condition sampling. Furthermore,

---

[2]We use the BLIP (Li et al., 2022) caption model to generate descriptions for HQ images as text prompts.

using visual embedding avoids the need for applying a complex text prompt that can describe LQ image in detail and accurately (Liao et al., 2024; Li et al., 2024b).

**Spatial Encoder.** However, the face-specific visual embedding $c_v$, while essential for capturing global semantic attributes, is insufficient for preserving global structure (① in Fig. 8). To address this issue, we introduce the Spatial Encoder (SE) to effectively extract and enhance spatial structure preservation (Fig. 3). We use the pretrained UNet encoder from stable diffusion to capture the full content of the LQ image, including structural information. When combined with the visual embedding, the SE then extracts the spatial features, denoted as $f_v = SE(x_l, c_v)$. The *ResNet* and *Attn* blocks represent the standard ResNet and Cross-Attention transformer blocks in LCM. The output from the *ResNet* block is used as the Query features, while the visual embedding $c_v$ serves as both Key and Value features in the *Attn* block. Then the spatial features is combined with the output of *Attn* block. After three iterations of LCM sampling, we finally generate the reconstructed HQ image $x_{rec} = \mathcal{D}(\boldsymbol{f_\theta}(z_{\tau_n}, c_v, \tau_n, f_v))$.

**Training Objectives.** To train the Visual Encoder and Spatial Encoder, we adopt three image-level losses: reconstruction loss $\mathcal{L}_1$, a perceptual loss (Johnson et al., 2016; Zhang et al., 2018) $\mathcal{L}_{per}$, and an adversarial loss (Goodfellow et al., 2014; Esser et al., 2021) $\mathcal{L}_{adv}$:

$$\mathcal{L}_1 = \|x_h - x_{rec}\|_1; \quad \mathcal{L}_{per} = \|\Phi(x_h) - \Phi(x_{rec})\|_2^2; \quad \mathcal{L}_{adv} = [\log D(x_h) + \log(1 - D(x_{rec}))],$$

where $x_h$ represents the HQ image, and $\Phi$ denotes the feature extractor of VGG19 (Simonyan & Zisserman, 2014). The complete objective function of our model is:

$$\mathcal{L} = \mathcal{L}_1 + \mathcal{L}_{per} + \lambda \mathcal{L}_{adv}, \tag{2}$$

where $\lambda$ is the trade-off parameter and set to 0.1 by default in the following experiments.

## 4 EXPERIMENTS

### 4.1 EVALUATION ON SYNTHETIC AND REAL-WORLD DATA

We evaluate our method on one *synthetic* dataset and three *real-world* datasets, which are commonly used for evaluation in blind face restoration tasks (Wang et al., 2021a; Zhou et al., 2022; Yue & Loy, 2024; Yang et al., 2024). We compare our method with recent baselines, including (CNN/Transformer-based methods) PULSE (Menon et al., 2020), DFDNet (Li et al., 2020), PS-FRGAN (Chen et al., 2021), GFPGAN (Wang et al., 2021a), GPEN (Yang et al., 2021), Restor-Former (Zamir et al., 2022), VQFR (Gu et al., 2022), CodeFormer (Zhou et al., 2022), (Diffusion-based methods) DR2 (Wang et al., 2023), DifFace (Yue & Loy, 2024), PGDiff (Yang et al., 2024), and WaveFace (Miao et al., 2024). See Appendix A for more details.

For the evaluation on the synthetic dataset (i.e., CelebA-Test (Karras et al., 2017)), we use five quantitative metrics: LPIPS (Zhang et al., 2018), FID (Heusel et al., 2017), MUSIQ, PSNR, and SSIM (Wang et al., 2004), similar to metrics used in CodeFormer (Zhou et al., 2022) and IDS used in VQFR (Gu et al., 2022) (also referred to as Deg). The results of the methods are summarized in Tab. 1 (the second to seventh columns). In terms of image quality metrics LPIPS and MUSIQ **(MUS.)**, our *InterLCM* achieves superior scores compared to existing methods. Furthermore, it faithfully preserves identity and structure, as evidenced by the best IDS and SSIM scores. Additionally, Fig. 6 demonstrates that our method significantly outperforms others, while the compared methods fail to yield satisfactory restoration results. For instance, DFDNet, PSFRGAN, GFPGAN, GPEN, DifFace, and PGDiff introduce noticeable artifacts, while PULSE and DR2 produce overly smoothed results that lack essential facial details. Moreover, while RestoreFormer, VQFR, and CodeFormer can generate high-quality texture details (e.g., *hair*), they still exhibit minor artifacts. In contrast, our method is slightly superior to theirs (see the zoomed-in area in Fig. 6).

For the evaluation on the real-world datasets (i.e., LFW-Test (Huang et al., 2008), WebPhoto-Test (Wang et al., 2021a), and WIDER-Test (Yang et al., 2016)), we adopt two quantitative metrics following the setting of CodeFormer (Zhou et al., 2022), namely FID and MUSIQ. The comparative results are summarized in Tab. 1 (the eight to thirteenth columns). We observe that our method achieves the best performance on WebPhoto-Test and WIDER-Test with medium and heavy degradation. In addition, it obtains the highest score in MUSIQ on the LFW-Test with mild degradation.

Table 1: Quantitative comparison on the *synthetic* and *real-world* dataset. The best results are in **bold**, and the second best results are underlined.

| Dataset | Synthetic dataset Celeba-Test | | | | | | Real-world datasets | | | | | | Time (Sec) |
|---|---|---|---|---|---|---|---|---|---|---|---|---|---|
| | | | | | | | LFW-Test | | WebPhoto-Test | | WIDER-Test | | |
| Method \ Metrics | LPIPS↓ | FID↓ | MUSIQ↑ | IDS↓ | PSNR↑ | SSIM↑ | FID↓ | MUSIQ↑ | FID↓ | MUSIQ↑ | FID↓ | MUSIQ↑ | |
| Input | 0.574 | 145.22 | 72.81 | 47.94 | 22.72 | 0.706 | 138.87 | 26.87 | 171.63 | 18.63 | 201.31 | 14.22 | – |
| PULSE | 0.356 | 68.33 | 66.46 | 43.98 | 22.10 | 0.592 | 67.01 | 65.00 | 85.69 | 63.88 | 70.65 | 63.01 | 3.509 |
| DFDNet | 0.332 | 54.21 | 72.08 | 40.44 | 24.27 | 0.628 | 60.28 | 73.06 | 92.71 | 68.50 | 59.56 | 62.02 | 0.438 |
| PSFRGAN | 0.294 | 54.21 | 73.32 | 39.63 | 24.66 | 0.661 | 49.89 | 73.60 | 85.42 | 71.67 | 85.42 | 71.50 | **0.041** |
| GFPGAN | 0.230 | 49.84 | 73.90 | 34.56 | 24.64 | 0.688 | 50.36 | 73.57 | 87.47 | 72.08 | 39.45 | 72.79 | 0.059 |
| GPEN | 0.290 | 63.44 | 67.52 | 36.17 | **25.48** | 0.708 | 61.04 | 68.96 | 99.09 | 61.10 | 46.25 | 62.64 | 0.109 |
| RestoreFormer | 0.241 | 50.04 | 73.85 | 36.16 | 24.61 | 0.660 | 48.77 | 73.70 | 78.85 | 69.83 | 50.04 | 67.83 | 0.066 |
| VQFR | 0.245 | 41.84 | 75.18 | 35.74 | 24.06 | 0.660 | 51.33 | 71.74 | 75.77 | 72.02 | 44.09 | 74.01 | 0.177 |
| CodeFormer | 0.227 | 52.94 | 75.55 | 37.27 | 25.15 | 0.685 | 52.84 | 75.48 | 83.95 | 74.00 | 39.22 | 73.41 | 0.085 |
| DR2 | 0.264 | 54.48 | 67.99 | 44.00 | 25.03 | 0.617 | 45.71 | 71.50 | 109.24 | 62.37 | 48.20 | 60.28 | 1.775 |
| DifFace | 0.272 | **39.23** | 68.87 | 45.80 | 24.80 | 0.684 | 46.31 | 69.76 | 80.86 | 65.37 | 37.74 | 65.02 | 3.248 |
| PGDiff | 0.300 | 47.26 | 71.81 | 55.90 | 22.72 | 0.659 | **44.65** | 71.74 | 101.68 | 67.92 | 38.38 | 68.26 | 14.768 |
| WaveFace | – | – | – | – | – | – | 53.88 | 73.54 | 78.01 | 70.45 | 37.23 | 72.89 | 19.370 |
| **Ours** | **0.223** | 45.38 | **76.58** | **33.64** | 25.19 | **0.718** | 51.32 | **76.16** | **75.48** | **75.88** | **35.43** | **76.29** | 0.421 |

(Left side labels: *CNN/Transformer-based* for PULSE through CodeFormer; *Diffusion-based* for DR2 through Ours.)

Figure 6: Qualitative comparisons of baselines on the synthetic of CelebA-Test for BFR (*Zoom in for a better view* and see Appendix F for additional results).

For the qualitative comparison in Fig. 7, we observe that our method demonstrates excellent robustness to real-world degradation, producing the most visually satisfactory results. Even in images with heavy degradation, our method generates rich texture details, whereas the compared methods exhibit noticeable artifacts. For example, as shown in Fig. 7 (the fifth and sixth rows), under heavy degradation in LQ image, all the compared methods produce face images with noticeable artifacts, whereas our method generates high-quality face images with rich hair details.

## 4.2 ABLATION STUDIES

**Effectiveness of Visual Encoder and Spatial Encoder.** Our proposed method starts from second step combining with both visual embedding from Visual Encoder (VE) and spatial features from Spatial Encoder (SE). We first evaluate the efficacy of visual embedding and spatial features, starting from second step, by exploring various ablated designs and comparing their performances. The ablated designs include: ① VE+2nd: The SE is removed, focusing only on VE training. ② Null-Text+SE+2nd: Only SE is trained, and VE is replaced by NullText. ③ Text+SE+2nd: Only SE is trained, and VE is replaced by Text. Performance results and comparion are presented in Fig. 8 (the first row, the second to fourth columns) and Tab. 2 (the first to third rows). We observe that ① VE+2nd captures the face-specific semantic information of the LQ image with high-quality detail, but is insufficient for preserving the global structure because visual embedding only provides semantically consistent content. ② NullText+SE+2nd and ③ Text+SD+2nd (e.g., "A photo of a human face" as shown in Fig. 8) receive spatial features that effectively capture the global facial structure of the LQ image; however, they compromise on detailed content (e.g., *eyes* and *wrinkles*).

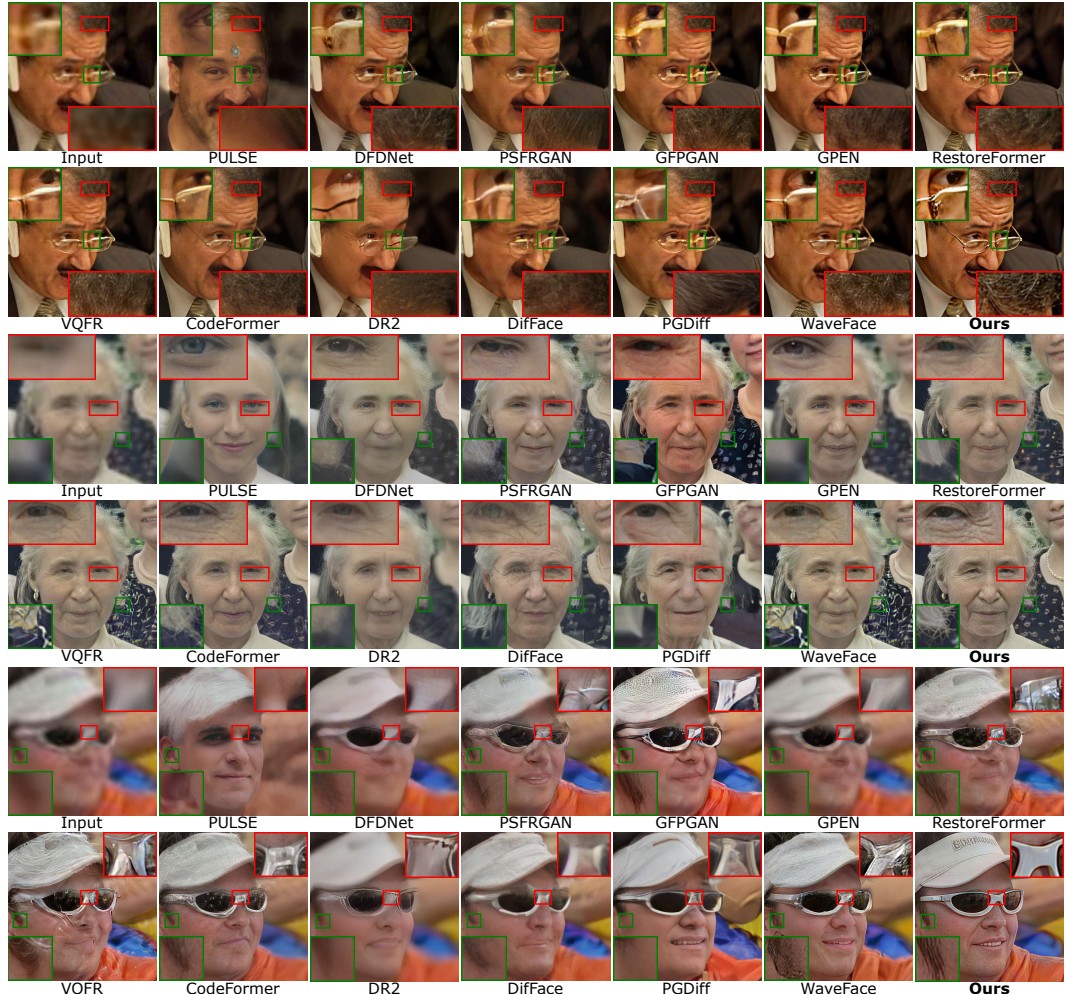

Figure 7: Qualitative comparisons of baselines on the real-world images from LFW-Test, WebPhoto-Test, and WIDER-Test (see Appendix F for additional results). **Zoom in for a better view**.

Table 2: Ablation study of Visual Encoder (VE) and Spatial Encoder (SE), as well as starting intermediate steps.

| Exp. | Text embedding | | | Starting steps | | | | LFW-Test | | WebPhoto-Test | | WIDER-Test | |
|---|---|---|---|---|---|---|---|---|---|---|---|---|---|
| | VE | Null Text | SE | 1st | 2nd | 3rd | 4th | FID↓ | MUS.↑ | FID↓ | MUS.↑ | FID↓ | MUS.↑ |
| ① | ✓ | | | | ✓ | | | 69.99 | 76.11 | 93.40 | 75.58 | 57.66 | 76.14 |
| ② | | ✓ | ✓ | | ✓ | | | 55.56 | 76.02 | 76.06 | 75.15 | 37.28 | 75.68 |
| ③ | | ✓ | ✓ | | ✓ | | | 55.07 | 75.75 | 77.76 | 75.30 | 36.15 | 75.98 |
| ④ | ✓ | | ✓ | ✓ | | | | 54.94 | 71.50 | 92.33 | 72.92 | 40.72 | 71.00 |
| ⑤ | ✓ | | ✓ | | | ✓ | | **50.48** | 75.06 | 86.53 | 73.66 | 38.71 | 73.18 |
| ⑥ | ✓ | | ✓ | | | | ✓ | 50.59 | 71.36 | 77.25 | 72.01 | 50.70 | 70.41 |
| ⑦‡ | ✓ | | ✓ | | ✓ | | | 51.32 | **76.16** | **75.48** | **75.88** | **35.43** | **76.29** |

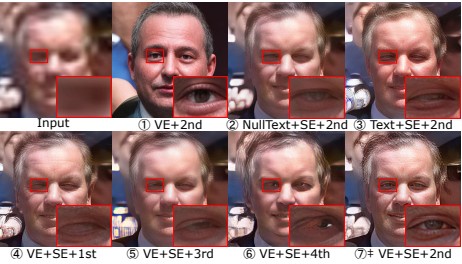

Figure 8: Visualization of the ablation study for various design variants. ‡ indicates our results.

We also experimentally confirm the starting step and present the results in Fig. 8 (the second row) and Tab. 2 (fourth to seventh rows). It can be observed that starting from the initial step (i.e., noise), as shown in ④ VE+SE+1st, generates detailed textures (e.g., *wrinkles*) but introduces randomness (e.g., *eyes*). Starting from a later step, ⑤ VE+SE+3rd and ⑥ VE+SE+4th result in blurred outputs (the second row, the second column in Fig. 8) and preserving the textures of the LQ image (the second row, the third column in Fig. 8) but fail to generate fine details, due to the limitations imposed by the number of denoising iterations. Thus, we incorporate both the visual embedding and spatial features into the LCM, starting from the second step, which facilitates the capture of face-specific information and the generation of fine details (⑦ in Fig. 8 and the last row in Tab. 2).

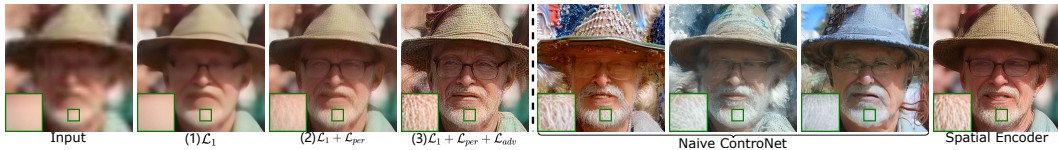

Figure 9: (*Left*) visualization of the ablation study for both the perceptual and adversarial losses. (*Right*) visualization of the ablation study comparing the naive ControlNet and our Spatial Encode.

Table 3: Ablation study of both the perceptual and adversarial losses.

| Exp. | $\mathcal{L}_1$ | $\mathcal{L}_{per}$ | $\mathcal{L}_{adv}$ | LFW-Test FID↓ | LFW-Test MUSIQ↑ | WebPhoto-Test FID↓ | WebPhoto-Test MUSIQ↑ | WIDER-Test FID↓ | WIDER-Test MUSIQ↑ |
|---|---|---|---|---|---|---|---|---|---|
| (a) | ✓ | | | 87.12 | 43.14 | 141.86 | 39.37 | 93.61 | 33.71 |
| (b) | ✓ | ✓ | | 57.57 | 67.99 | 95.02 | 66.24 | 44.83 | 63.94 |
| (c) Ours | ✓ | ✓ | ✓ | 51.32 | 76.16 | 75.48 | 75.88 | 35.43 | 76.29 |

Table 4: Ablation study of the naive ControlNet and our proposed Spatial Encoder.

| Exp. | Loss | LFW-Test FID↓ | LFW-Test MUSIQ↑ | WebPhoto-Test FID↓ | WebPhoto-Test MUSIQ↑ | WiDER-Test FID↓ | WiDER-Test MUSIQ↑ |
|---|---|---|---|---|---|---|---|
| Naive ControlNet | Eq. (1) | **35.43** | 75.03 | 81.91 | 73.63 | 49.58 | 74.20 |
| Spatial Encoder | Eq. (2) | 55.07 | **75.75** | **77.76** | **75.30** | **36.15** | **75.98** |

**Inference time.** Tab. 1 (the last column) shows the inference time of different methods. All methods are evaluated on input images using a Quadro RTX 3090 GPU (24GB VRAM) with resolution of $512 \times 512$. The sampling time of our method has a similar running time as CNN/Transformer-based methods, such as DFDNet (Li et al., 2020), GPEN (Yang et al., 2021), and VQFR (Gu et al., 2022). Meanwhile, the inference time of our method significant surpass that of other diffusion-based methods, such as PGDiff (Yang et al., 2024) and WaveFace (Miao et al., 2024), which remain constrained by the iterative sampling processes inherent to diffusion models.

**Effectiveness of perceptual and adversarial losses.** We consider that the superior restoration performance of our *InterLCM* is mainly due to the integrating with both perceptual loss (Johnson et al., 2016) and adversarial loss (Goodfellow et al., 2014) in the image domain, which are commonly used in restoration model training leading to a high-quality and high-fidelity face restoration output. To highlight the effectiveness of these two losses, we perform the ablation experiments in Fig. 9 (Left) and Tab. 3. We can see that without perceptual and adversarial losses, the quantitative metrics are significantly degraded (Tab. 3 (the first row)), as it is challenging to achieve good visual quality using only reconstruction loss (Fig. 9 (the second column)). Adding perceptual loss and adversarial loss in the image domain can effectively restore realistic details. In addition, we also conduct an ablation study on Spatial Encoder in *InterLCM* and Naive ControlNet (Fig. 9 (Right) and Tab. 4). The primary difference between the two lies in the loss function utilized during training. Although Naive ControlNet can generate high-quality image while maintaining structure, it loses fidelity due to the denoising loss focuses on the semantic information but fidelity (Zhang et al., 2023).

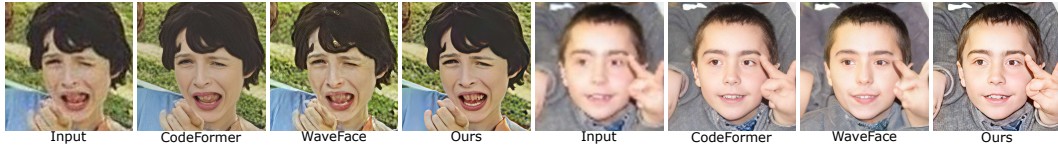

Figure 10: Input LQ images with hands may experience failing restorations.

## 5 CONCLUSION

In this paper, we proposed *InterLCM*, a novel framework for blind face restoration (BFR) that leverages the latent consistency model (LCM) to improve semantic consistency and restore high-quality images from low-quality inputs. By treating the low-quality image as *an intermediate step in the LCM*, *InterLCM* achieves more accurate restorations with fewer sampling steps compared to traditional diffusion-based methods. Additionally, we integrated a CLIP-based image encoder and visual encoder to capture face-specific semantic information and a spatial encoder based on ControlNet to ensure structural consistency. Extensive experiments on both synthetic and real-world datasets demonstrated that *InterLCM* outperforms existing approaches, delivering superior image quality and faster inference, particularly in challenging real-world scenarios with unpredictable degradations.

**Limitation.** Although our method excels in the existing methods in blind face restoration, it does not depart from limitations. When *InterLCM* deals with images that include hands, it excels at generating more facial details but does not produce realistic hands (Fig. 10). That probably results from the fact that the FFHQ training dataset contains a very limited number of such images. One potential solution is to enhance the training data by adding more diverse face images with hands.

ACKNOWLEDGEMENTS

This work was supported by NSFC (NO. 62225604) and Youth Foundation (62202243). We acknowledge the support of the project PID2022-143257NB-I00, funded by the Spanish Government through MCIN/AEI/10.13039/501100011033 and FEDER. We acknowledge "Science and Technology Yongjiang 2035" key technology breakthrough plan project (2024Z120). Computation is supported by the Supercomputing Center of Nankai University (NKSC).

We gratefully acknowledge Chongyi Li, Professor at Nankai University, China, for his valuable discussions and comments.

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

APPENDIX

# A APPENDIX: IMPLEMENTATION DETAILS

## A.1 TRAINING DETAILS

We mainly use the pre-trained LCM, distilled from StableDiffusion 1.5. The Spatial Encoder is partially initialized using UNet encoder from the pre-trained Stable Diffusion 1.5, following the approach in (Zhang et al., 2023). The decoder from CodeFormer (Zhou et al., 2022) serves as the Visual Encoder, with adjustments made to the input and output dimensions to align with our settings. The proposed method is implemented in Pytorch (Paszke et al., 2017). We use Adam (Kingma & Ba, 2015) with a batch size 8, using a learning rate of $2 \times 10^{-5}$. The models are trained for 15K iterations using eight A40 GPUs (48GB VRAM).

## A.2 TRAINING DATA

We train our models on the FFHQ dataset (Karras et al., 2019), which consists of 70,000 HQ face images with a resolution of $1024 \times 1024$. First, we resize the HQ images to $512 \times 512$. The resized images are then degraded to generate LQ images following the typical degradation process described in (Zhou et al., 2022):

$$x_l = \{[(x_h * k_\sigma)\downarrow_s + n_\delta]_{\text{JPEG}_q}\}\uparrow_s, \tag{3}$$

where $x_h$ and $x_l$ represent the HQ and LQ images, respectively, $k_\sigma$ is the Gaussian kernel with $\sigma \in \{1 : 15\}$, $\downarrow_s$ represents the downsampling operation with a scale factor $s \in \{1 : 30\}$, and $n_\delta$ denotes Gaussian noise with a standard deviation of $\delta \in \{0 : 20\}$. The convolution operation is denoted by $*$, followed by JPEG compression with a quality factor of $q \in \{30 : 90\}$. Finally, an upsampling operation $\uparrow_s$ with scale $s$ is applied to restore the original resolution of $512 \times 512$.

## A.3 TEST DATA.

We evaluate our method on one *synthetic* dataset and three *real-world* datasets, which are commonly used for evaluation in blind face restoration tasks (Wang et al., 2021a; Zhou et al., 2022; Yue & Loy, 2024; Yang et al., 2024). The synthetic dataset, CelebA-Test (Karras et al., 2017), contains 4,000 high-quality (HQ) images. The corresponding low-quality (LQ) images are synthesized using the same degradation process as described in Eq. (3), which is consistent with our training setting. The three real-world datasets encompass varying degrees of degradation: LFW-Test (Huang et al., 2008) with mild, WebPhoto-Test (Wang et al., 2021a) with medium, and WIDER-Test (Yang et al., 2016) with heavy degradation. They contain 1,711, 407, and 970 LQ images, respectively.

## A.4 BASELINE IMPLEMENTATIONS.

We compare our method with recent baselines, including (CNN/Transformer-based methods) PULSE (Menon et al., 2020) [3], DFDNet (Li et al., 2020) [4], PSFRGAN (Chen et al., 2021) [5], GFPGAN (Wang et al., 2021a) [6], GPEN (Yang et al., 2021) [7], RestorFormer (Zamir et al., 2022) [8], VQFR (Gu et al., 2022) [9], CodeFormer (Zhou et al., 2022) [10], (Diffusion-based methods) DR2 (Wang et al., 2023) [11], DifFace (Yue & Loy, 2024) [12], PGDiff (Yang et al., 2024) [13], and

---

[3] https://github.com/krantirk/Self-Supervised-photo
[4] https://github.com/csxmli2016/DFDNet
[5] https://github.com/chaofengc/PSFRGAN
[6] https://github.com/TencentARC/GFPGAN
[7] https://github.com/yangxy/GPEN
[8] https://github.com/swz30/Restormer
[9] https://github.com/TencentARC/VQFR
[10] https://github.com/sczhou/CodeFormer
[11] https://github.com/Kaldwin0106/DR2_Drgradation_Remover
[12] https://github.com/zsyOAOA/DifFace
[13] https://github.com/pq-yang/PGDiff

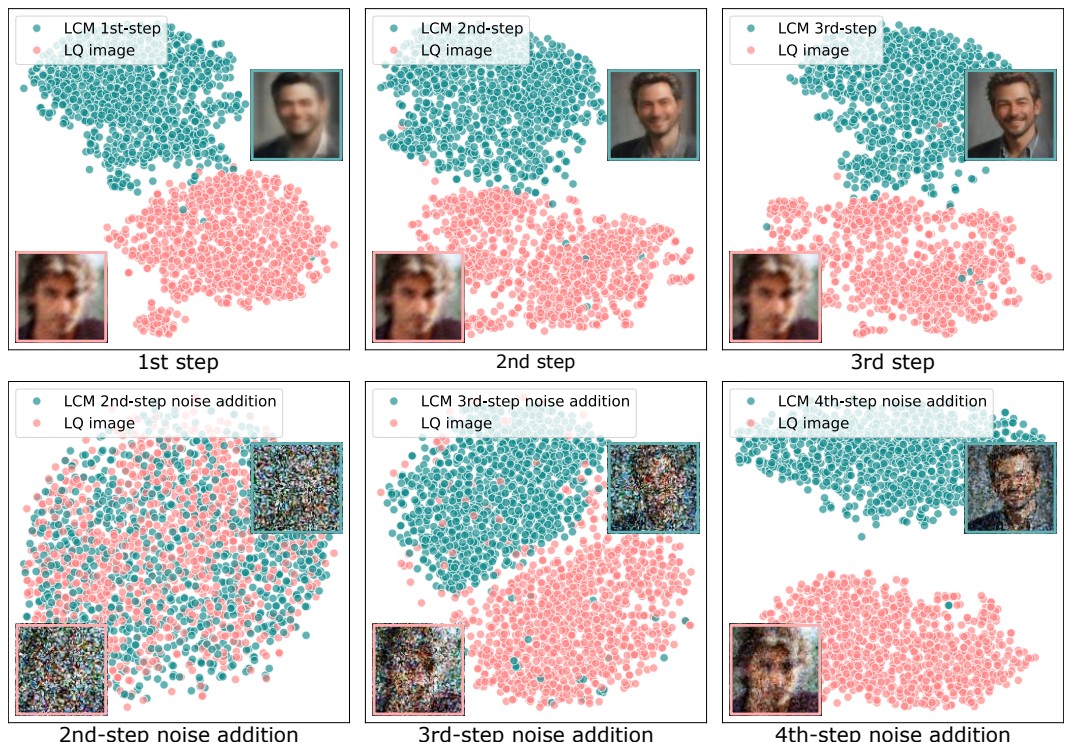

Figure 11: t-SNE (Hinton & Roweis, 2002) visualizations of feature distributions show (Left) the first step sampling result of LCM and the LQ image (FID=103.70) with their noise-added versions (FID=2.83); (Middle) the second step result and the LQ image (FID=157.80) with their noise-added versions (FID=31.83); (Right) the third step result and the LQ image (FID=172.66) with their noise-added versions (FID=214.40).

WaveFace (Miao et al., 2024) [14]. The evaluation of all methods was conducted on images with a resolution of $512 \times 512$, utilizing their publicly available official code and default settings.

# B  APPENDIX: ALGORITHM DETAIL OF *InterLCM*

---

**Algorithm 1** The sampling of *InterLCM*

---

**Input:** The LQ image $x_l$, Pretrained Latent Consistency Model combining with visual embedding from Visual Module and spatial features from Spatial Encoder (SE): $f_{\boldsymbol{\theta}}(z_{\tau_n}, c_v, \tau_n, f_v)$. Sequence of timesteps $\tau_1 > \tau_2 > \cdots > \tau_{N-1}, N = 4$. Noise schedule $\alpha(t), \sigma(t)$, Encoder $\mathcal{E}$, and Decoder $\mathcal{D}$.
Initial latent code $z_0 \leftarrow \mathcal{E}(x_l)$
**for** $n = 1$ to $N - 1$ **do**
    $z_{\tau_n} \sim \mathcal{N}(\alpha(\tau_n)z_0; \sigma^2(\tau_n)\mathbf{I})$
    $z_0 \leftarrow f_{\boldsymbol{\theta}}(z_{\tau_n}, c_v, \tau_n, f_v)$
**end for**
$x_{rec} \leftarrow \mathcal{D}(z_0)$
**Output:** $x_{rec}$

---

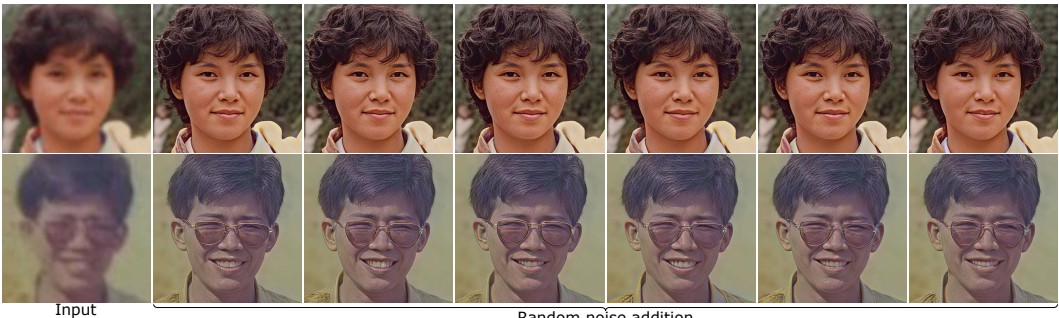

Figure 12: Two restoration examples of our *InterLCM* on the real-world dataset WebPhoto-Test, achieved through random noise addition in the three noise addition step of 4-step LCM.

## C  APPENDIX: ABLATION ANALYSIS

### C.1  SHOULD WE START FROM THE 2ND, 3RD, OR 4TH STEP IN THE LCM?

To leverage the content consistency inherent in LCM (Luo et al., 2023a), we retain the pretrained model and follow its sampling process. As shown in Fig. 2 (right, the first row), the 4-step LCM sampling process generates semantic consistency images. In the first step, LCM directly predicts an image from random noise. In subsequent steps, LCM first adds noise to the previous image and then predicts a finer output. In Fig. 11 (the first row), we visualize the feature distributions for the LQ image and the results of the first three sampling steps using t-SNE (Hinton & Roweis, 2002). We can observe that the clusters are well-separable (Fig. 11 (the first row)). Based on the three addition processes in each of the 4-step LCM, we move the LQ image to each intermediate state of LCM (Fig. 11 (the second row)). We find that the distribution of the LQ image is closest to that of the generated image after the first noise addition (second step noise addition) than other intermediate states (Fig. 11 (the second row, the first column)). Therefore, we use the LQ image as the intermediate state after the first noise addition in LCM. Subsequently, the LCM is applied starting from *the second step*.

### C.2  ROBUSTNESS TO RANDOM NOISE ADDITION

As shown in Fig. 12, we showcase our robustness to random noise addition in the three noise addition step of 4-step LCM. Our $InterLCM$ effectively restores the face-specific detail using random noise addition.

## D  APPENDIX: ABLATION ANALYSIS

### D.1  EFFECTIVENESS OF PERCEPTUAL AND ADVERSARIAL LOSSES

We conducted an ablation study by removing the perceptual loss while retaining the adversarial loss. As shown in Tab. 5, we can see that without perceptual loss (3), the quantitative metrics are significantly degraded, as it is challenging to reconstruct the textual detail using adversarial loss (see Fig. 13). This indicates that the perceptual loss plays a crucial role in the fidelity of the restored faces.

### D.2  OUR METHOD USING LCM IN DIFFERENT NUMBERS OF STEPS

LCM employs a 4-step inference process to balance image quality and inference time, as recommended by the original paper (Luo et al., 2023a). In this paper, we use the recommended 4-step LCM model, while we also offer an ablation study with a 2-step LCM model. As observed from the Tab. 6, the 4-step LCM only works slightly worse than the 2-step LCM on two metrics, which is utilized as the backbone for our *InterLCM*.

---

[14]https://github.com/yoqim/waveface

Table 5: Ablation study of both the perceptual and adversarial losses. The best results are shown in **bold**.

| Dataset | | | | *Synthetic* dataset | | | | | | *Real-world* datasets | | | | | |
|---|---|---|---|---|---|---|---|---|---|---|---|---|---|---|---|
| | | | | Celeba-Test | | | | | | LFW-Test | | WebPhoto-Test | | WIDER-Test | |
| Exp. | $\mathcal{L}_1$ | $\mathcal{L}_{per}$ | $\mathcal{L}_{adv}$ | LPIPS↓ | FID↓ | MUSIQ↑ | IDS↓ | PSNR↑ | SSIM↑ | FID↓ | MUSIQ↑ | FID↓ | MUSIQ↑ | FID↓ | MUSIQ↑ |
| (1) | ✓ | | | 0.403 | 93.36 | 41.68 | 35.48 | **27.37** | **0.764** | 87.12 | 43.14 | 141.86 | 39.37 | 93.61 | 33.71 |
| (2) | ✓ | ✓ | | 0.226 | 51.47 | 68.49 | **32.47** | 26.63 | 0.732 | 57.57 | 67.99 | 95.02 | 66.24 | 44.83 | 63.94 |
| (3) | ✓ | | ✓ | 0.369 | 72.29 | 76.01 | 36.03 | 24.67 | 0.653 | 63.21 | 76.09 | 129.21 | 75.62 | 100.15 | 74.40 |
| (4) Ours | ✓ | ✓ | ✓ | **0.223** | **45.38** | **76.58** | 33.64 | 25.19 | 0.718 | **51.32** | **76.16** | **75.48** | **75.88** | **35.43** | **76.29** |

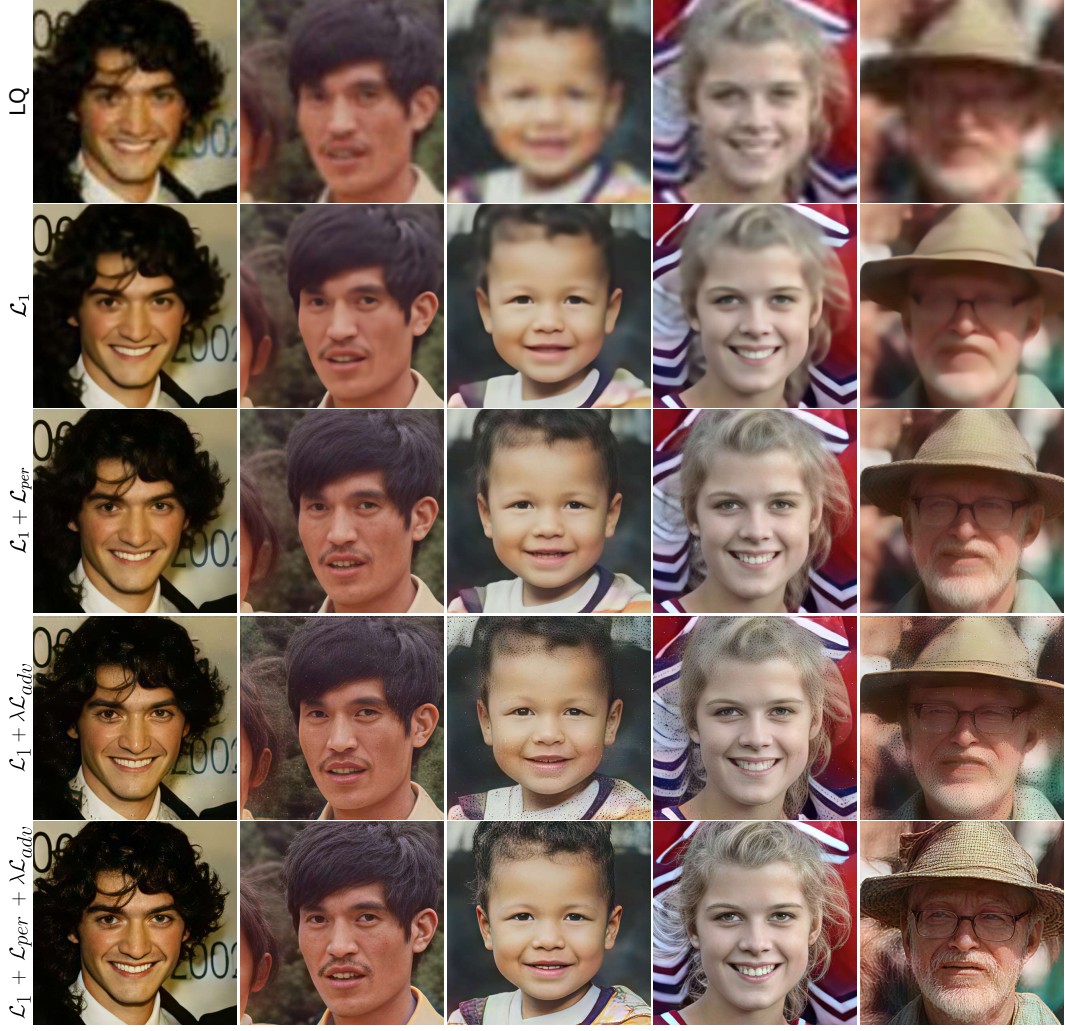

Figure 13: Visualization of the ablation study for both the perceptual and adversarial losses.

## D.3 THE ANALYSIS OF OUR METHOD USING LCM OR SD TURBO

We test the SD Turbo (Sauer et al., 2023) as the backbone to develop our method for the BFR problem. By the quantitative comparison in Tab. 7, we show that consistency model, which directly predict the $x_0$ in each step, better suits the BFR problem.

Table 6: Quantitative comparison using the LCM model with different inference steps. The best results are shown in **bold**.

| Dataset | *Synthetic* dataset | | | | | | *Real-world* datasets | | | | | |
|---|---|---|---|---|---|---|---|---|---|---|---|---|
| | Celeba-Test | | | | | | LFW-Test | | WebPhoto-Test | | WIDER-Test | |
| Metrics ⟍ Method ⟍ | LPIPS↓ | FID↓ | MUSIQ↑ | IDS↓ | PSNR↑ | SSIM↑ | FID↓ | MUSIQ↑ | FID↓ | MUSIQ↑ | FID↓ | MUSIQ↑ |
| Input | 0.574 | 145.22 | 72.81 | 47.94 | 22.72 | 0.706 | 138.87 | 26.87 | 171.63 | 18.63 | 201.31 | 14.22 |
| Ours (2-step LCM) | 0.248 | 49.19 | 74.31 | 34.92 | 23.91 | 0.662 | 56.21 | **76.24** | 75.84 | **76.11** | 38.23 | 76.00 |
| Ours (4-step LCM) | **0.223** | **45.38** | **76.58** | **33.64** | **25.19** | **0.718** | **51.32** | 76.16 | **75.48** | 75.88 | **35.43** | **76.29** |

Table 7: Quantitative comparison with SD Turbo or LCM as the backbones for the blind face restoration (BFR) model. The best results are in **bold**.

| Dataset | *Synthetic* dataset | | | | | | *Real-world* datasets | | | | | |
|---|---|---|---|---|---|---|---|---|---|---|---|---|
| | Celeba-Test | | | | | | LFW-Test | | WebPhoto-Test | | WIDER-Test | |
| Metrics ⟍ Method ⟍ | LPIPS↓ | FID↓ | MUSIQ↑ | IDS↓ | PSNR↑ | SSIM↑ | FID↓ | MUSIQ↑ | FID↓ | MUSIQ↑ | FID↓ | MUSIQ↑ |
| Input | 0.574 | 145.22 | 72.81 | 47.94 | 22.72 | 0.706 | 138.87 | 26.87 | 171.63 | 18.63 | 201.31 | 14.22 |
| Ours (SD Turbo) | 0.257 | 48.51 | 74.15 | 37.02 | 23.30 | 0.660 | 56.44 | 74.24 | 84.66 | 74.41 | 43.53 | 72.35 |
| Ours (LCM) | **0.223** | **45.38** | **76.58** | **33.64** | **25.19** | **0.718** | **51.32** | **76.16** | **75.48** | **75.88** | **35.43** | **76.29** |

## D.4    OUR METHOD USING LCM-LoRA

We test the LCM-LoRA (Luo et al., 2023b) as the backbone to develop our method for the BFR problem. Tab. 8 and Fig. 14 show the qualitative and quantitative, respectively, comparison of using LCM-LoRA and our method. As shown in Tab. 8, LCM-LoRA does not perform as well as our method in terms of LPIPS and FID metrics, while it achieves better results on the MUSIQ metric for image quality evaluation on real datasets, such as LFW-Test and WebPhoto-Test. The qualitative results in Fig. 14 demonstrate that both LCM-LoRA and our method can achieve high-quality reconstructed images.

## D.5    OUR METHOD USING ONE-STEP MODELS ($x_0$-PREDICTION-BASED DIFFUSION MODELS)

We use one-step models ($x_0$-prediction-based diffusion models) as the backbone to develop our method for the BFR task. We first move the LQ image to the noise space of the one-step models. We make some comparisons in Tab. 9 and Fig. 15. As shown in Table Tab. 9, our metrics significantly outperform one-step diffusion models in the BFR task, except for the FID metric on the Synthetic dataset. As shown in the qualitative comparison in Fig. 15, results of our method using one-step models (as shown in the second and third rows) indicate that these models face challenges with artifacts and blur when reconstructing high-quality images, while our method can reconstruct high-quality images with detailed textures (the fourth row).

## E    APPENDIX: ADDITIONAL ANALYSIS

### E.1    CAN WE REGARD THE LQ IMAGE AS AN INTERMEDIATE RESULT IN SD SAMPLING?

When we perform SD sampling, the Gaussian noise $z_T$ is gradually denoised into a clear image $z_0$ (see Fig. 16). We use the DDIM schedule with $T = 50$. The intermediate result of SD sampling lacks a lot of image detail, while the LQ image mainly loses texture detail compared to the HQ image. Intuitively, we regard the LQ image as an intermediate result of SD sampling, especially at small noise levels (see Fig. 16 (red box)). As shown in Fig. 17, we regard the LQ image as the intermediate result at timesteps $t = 10, 20$, and 30 (the second column to fourth columns) and perform the remaining steps of the SD sampling, both for real-world LQ image (the first row) and synthetic LQ image (the second row). When we regard the LQ image as the intermediate

Table 8: Quantitative comparison with LCM-LoRA or LCM as the backbones for the blind face restoration (BFR) model. The best results are in **bold**.

| Dataset | Synthetic dataset | | | | | | Real-world datasets | | | | | |
| --- | --- | --- | --- | --- | --- | --- | --- | --- | --- | --- | --- | --- |
| | Celeba-Test | | | | | | LFW-Test | | WebPhoto-Test | | WIDER-Test | |
| Metrics / Method | LPIPS↓ | FID↓ | MUSIQ↑ | IDS↓ | PSNR↑ | SSIM↑ | FID↓ | MUSIQ↑ | FID↓ | MUSIQ↑ | FID↓ | MUSIQ↑ |
| Input | 0.574 | 145.22 | 72.81 | 47.94 | 22.72 | 0.706 | 138.87 | 26.87 | 171.63 | 18.63 | 201.31 | 14.22 |
| Ours (LCM-LoRA) | 0.240 | 53.26 | **76.58** | 35.48 | 24.14 | 0.661 | 54.70 | **76.26** | 82.08 | **76.59** | 39.62 | 75.81 |
| Ours (LCM) | **0.223** | **45.38** | **76.58** | 33.64 | 25.19 | **0.718** | 51.32 | 76.16 | **75.48** | 75.88 | **35.43** | **76.29** |

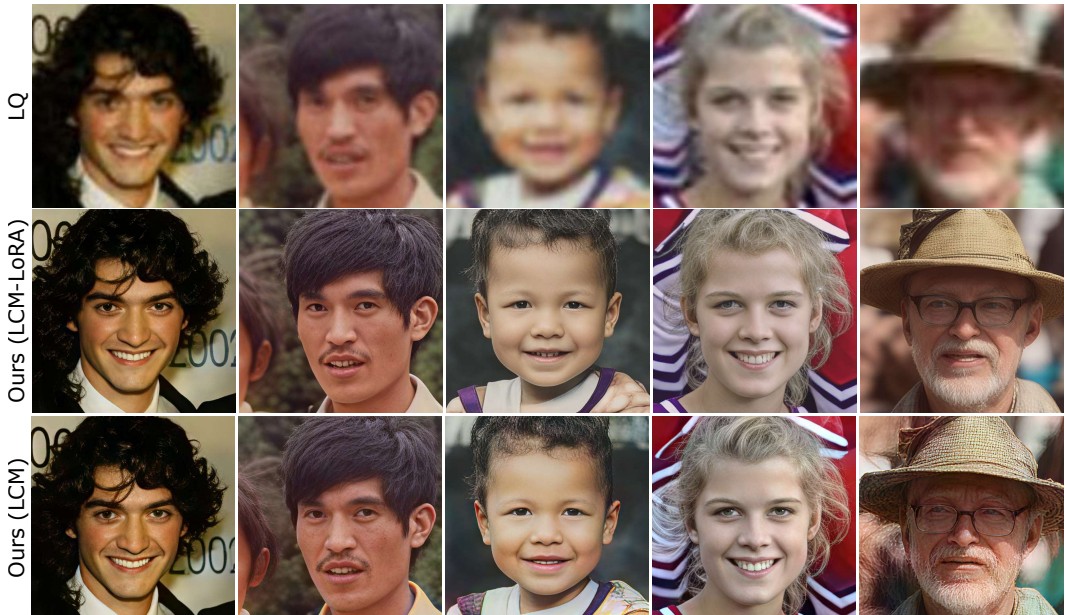

Figure 14: Results using LCM-LoRA and LCM backbone for our method.

result with small noise levels, the remaining SD denoise process tends to remove the potential noise in the LQ image. However, this process does not aid in image restoration but instead makes the image smoother (Fig. 17 (the second column)). Moreover, when we perform the SD denoise process starting with a high noise level using the LQ image, more edge information, such as details of glasses, can be lost (Fig. 17 (the third to fourth columns)). In conclusion, the degradation of the LQ image is different from that of the noised image at the intermediate step of SD sampling, even at small noise levels.

### E.2 CAN WE USE SUPER-RESOLUTION METHODS FOR FACE RESTORATION?

The purpose of image super-resolution is to increase the resolution of an image while preserving its content and details as much as possible. In contrast, face restoration does not aim to increase image resolution but focuses on recovering image details from the same LQ resolution. As shown in Fig. 18, we naively attempt to use state-of-the-art super-resolution methods (Rombach et al., 2022; Wang et al., 2024b; 2021b) to perform face restoration (the second to fourth columns). We first downsample an LQ image from a resolution of 512 to 128, then use it as the input for the super-resolution method to generate an image with a resolution of 512 (the second to fourth columns). The downsampled image at 128 resolution is upsampled to 512 resolution using bicubic interpolation (Fig. 18 (the first column)), and this upsampled image is then used as input for our method to produce the restored image (Fig. 18 (the last column)).

Table 9: Our method using one-step models ($x_0$-prediction-based diffusion models), The best results are in **bold**.

| Dataset | *Synthetic* dataset | | | | | | *Real-world* datasets | | | | | |
|---|---|---|---|---|---|---|---|---|---|---|---|---|
| | Celeba-Test | | | | | | LFW-Test | | WebPhoto-Test | | WIDER-Test | |
| Method / Metrics | LPIPS↓ | FID↓ | MUSIQ↑ | IDS↓ | PSNR↑ | SSIM↑ | FID↓ | MUSIQ↑ | FID↓ | MUSIQ↑ | FID↓ | MUSIQ↑ |
| Input | 0.574 | 145.22 | 72.81 | 47.94 | 22.72 | 0.706 | 138.87 | 26.87 | 171.63 | 18.63 | 201.31 | 14.22 |
| Ours (1-step SD Turbo) | 0.273 | **36.87** | 74.00 | 37.82 | 24.89 | 0.658 | 61.21 | 70.24 | 87.77 | 70.47 | 54.45 | 71.51 |
| Ours (1-step LCM) | 0.240 | 46.66 | 74.06 | 37.45 | 24.66 | 0.697 | 55.72 | 73.45 | 89.90 | 72.41 | 37.16 | 70.45 |
| Ours (4-LCM) | **0.223** | 45.38 | **76.58** | **33.64** | **25.19** | **0.718** | **51.32** | **76.16** | **75.48** | **75.88** | **35.43** | **76.29** |

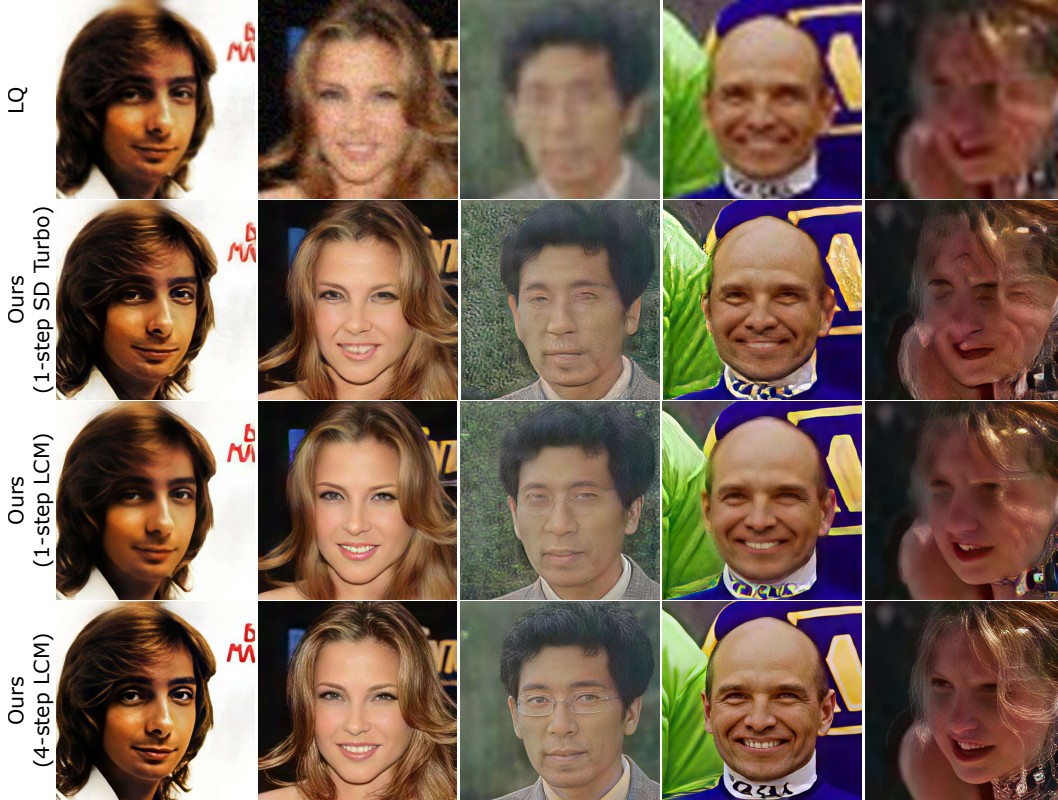

Figure 15: Results of our method using one-step models (as shown in the second and third rows) indicate that these models face challenges with artifacts and blur when reconstructing high-quality images, while our method can reconstruct high-quality images with detailed textures (the fourth row).

As shown in Fig. 18, the super-resolution methods struggle to recover facial details, whether applied to real-world or synthetic LQ images (the second to fourth columns). Although StableSR (Wang et al., 2024b) adds an additional 5,000 face images from the FFHQ dataset (Karras et al., 2019), it still struggles to recover facial details, such as hair and facial texture (the third column).

### E.3 ADDITIONAL RESULTS WITH TATTOOS OR FESTIVAL-STYLE FACE PAINT

As shown in Fig. 19 (the third and fourth rows), our method, *InterLCM*, demonstrates the ability to reconstruct high-quality details even in challenging cases, such as images featuring tattoos or festival-style face paint. However, when tattoos contain intricate details, such as text (e.g., the last column), accurately recovering these ambiguous elements during high-quality face reconstruction

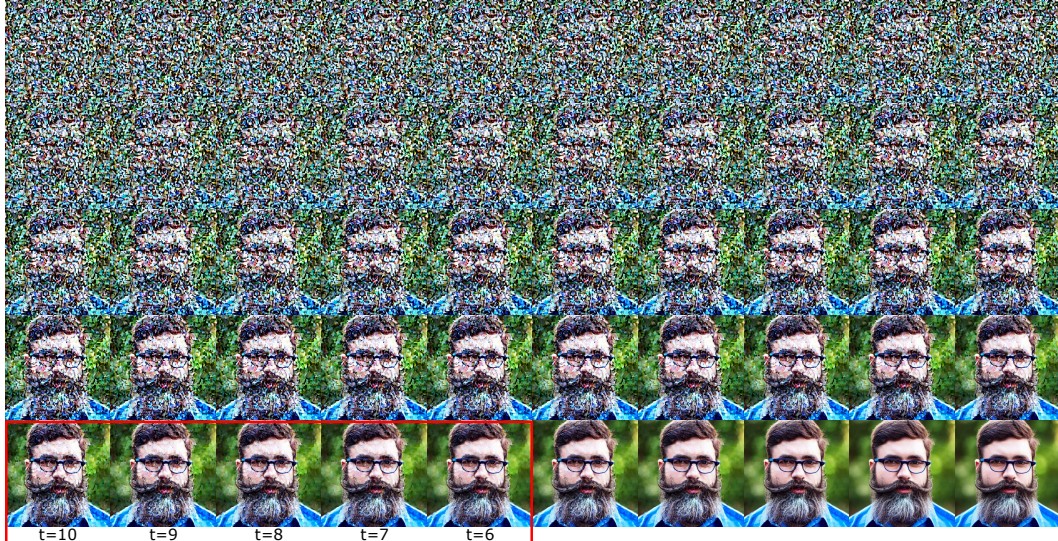

Figure 16: The generated results at each timestep of the diffusion sampling process from $T$ to 1. For example, given one prompt case "A man with a beard wearing glasses in blue shirt", the noise in the image is gradually reduced from timestep $T$ to 1, and the image is eventually generated with clarity (from left to right, top to bottom).

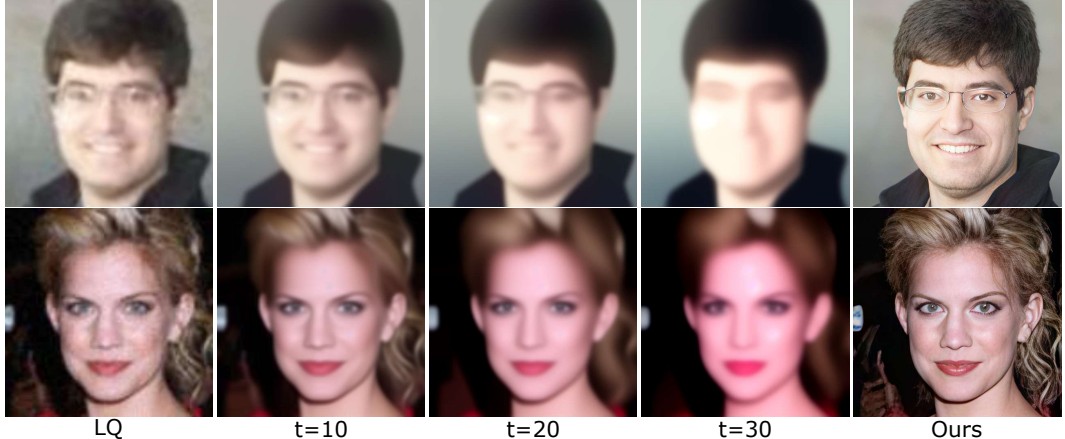

Figure 17: The generated results are obtained when we regard the LQ image as the intermediate result and perform the remaining steps of the SD sampling. For more detail, we regard the LQ image as the intermediate result at timesteps $t = 10, 20$, and 30 (the second column to fourth column), both for real-world LQ image (the first row) and synthetic LQ image (the second row). In these two examples, We use the prompts "A man with black hair wearing glasses in a black shirt" and "A woman with curly yellow hair", respectively

becomes challenging. This limitation may stem from the scarcity of such textures in the training dataset. An illustration of the complex textures in our training dataset FFHQ (Karras et al., 2019) is also shown in Fig. 19 (the first and second rows), where the festival-style face paints and rich-color hair appear multiple times during training.

### E.4 LQ SEMANTIC INFORMATION SUFFICES FOR HQ RECONSTRUCTION

In our method (Fig. 20, top(a)), *InterLCM*, we utilize a Visual Module to extract semantic information from LQ images for HQ reconstruction. To demonstrate that the LQ image suffices to provide

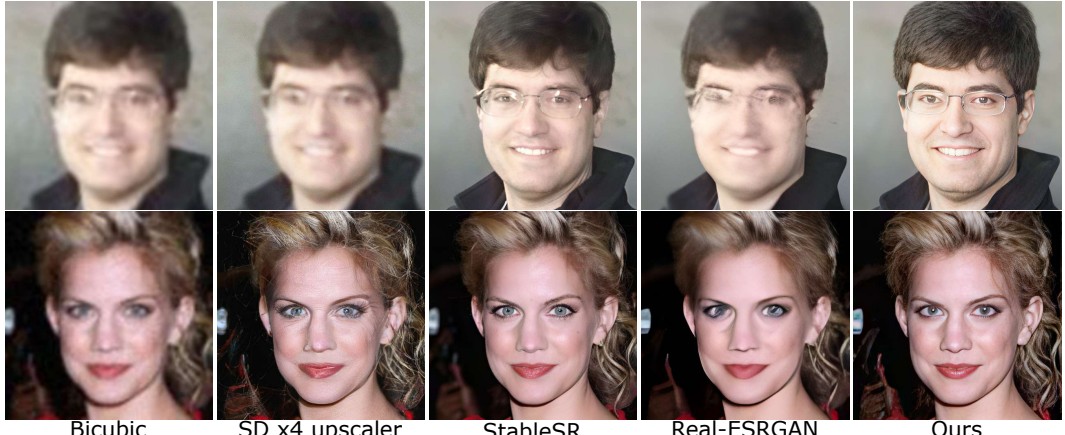

Bicubic     SD x4 upscaler     StableSR     Real-ESRGAN     Ours

Figure 18: The super-resolution methods struggle to recover facial details (the second to fourth columns).

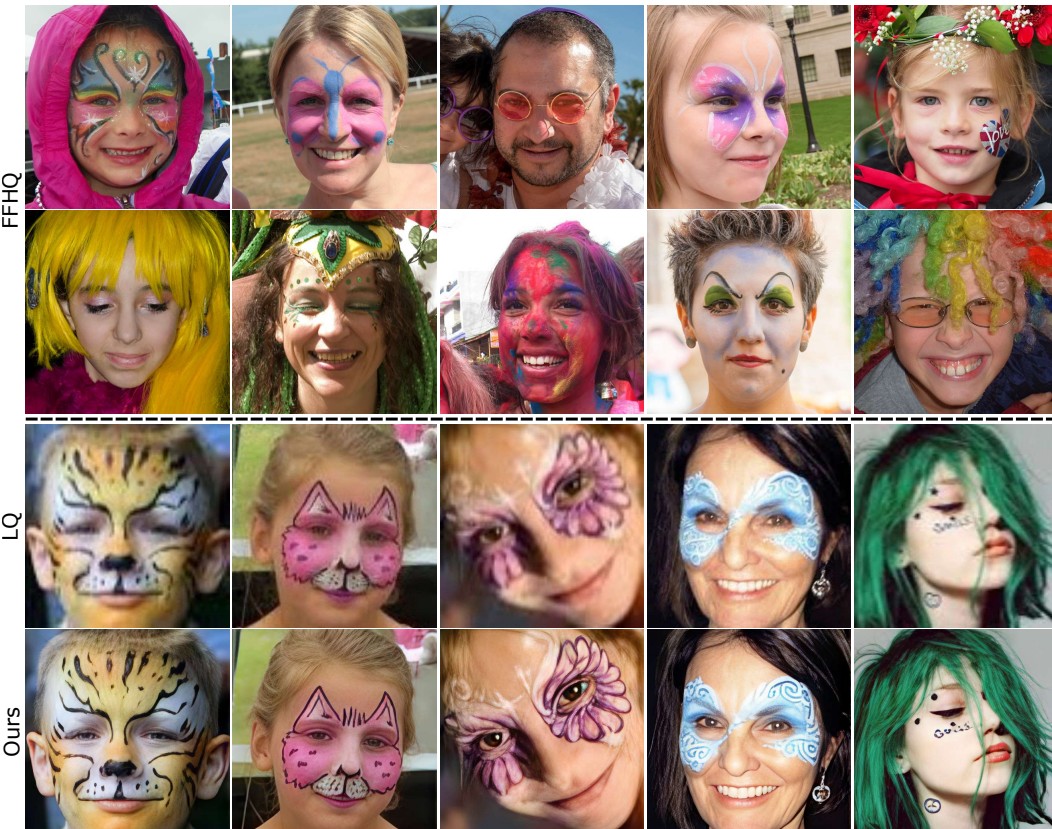

Figure 19: (top) In our training dataset FFHQ, there exist images containing festival-style face paint, as well as rich colors in the hair and head accessories. (bottom) Our method *InterLCM* can restore high-quality details for complex images with tattoos or festival-style face paint.

a prior for HQ reconstruction, we provide our model with LQ images exhibiting varying levels of degradation, decreasing from left to right (Fig. 20, middle(the first row)). The reconstruction results (Fig. 20, middle(the second row)) show that the semantic information from the LQ image suffices as a prior for HQ reconstruction when the degradation level is below a specific threshold (Fig. 20, middle(the third to fifth columns)).

Meanwhile, we observe that when the HQ image is used as the input to both the Visual Module and Spatial Encoder (Fig. 20, top(b)), the reconstructed image displays similar semantic information to that obtained using the LQ image (Fig. 20, bottom(the first column)). This result further

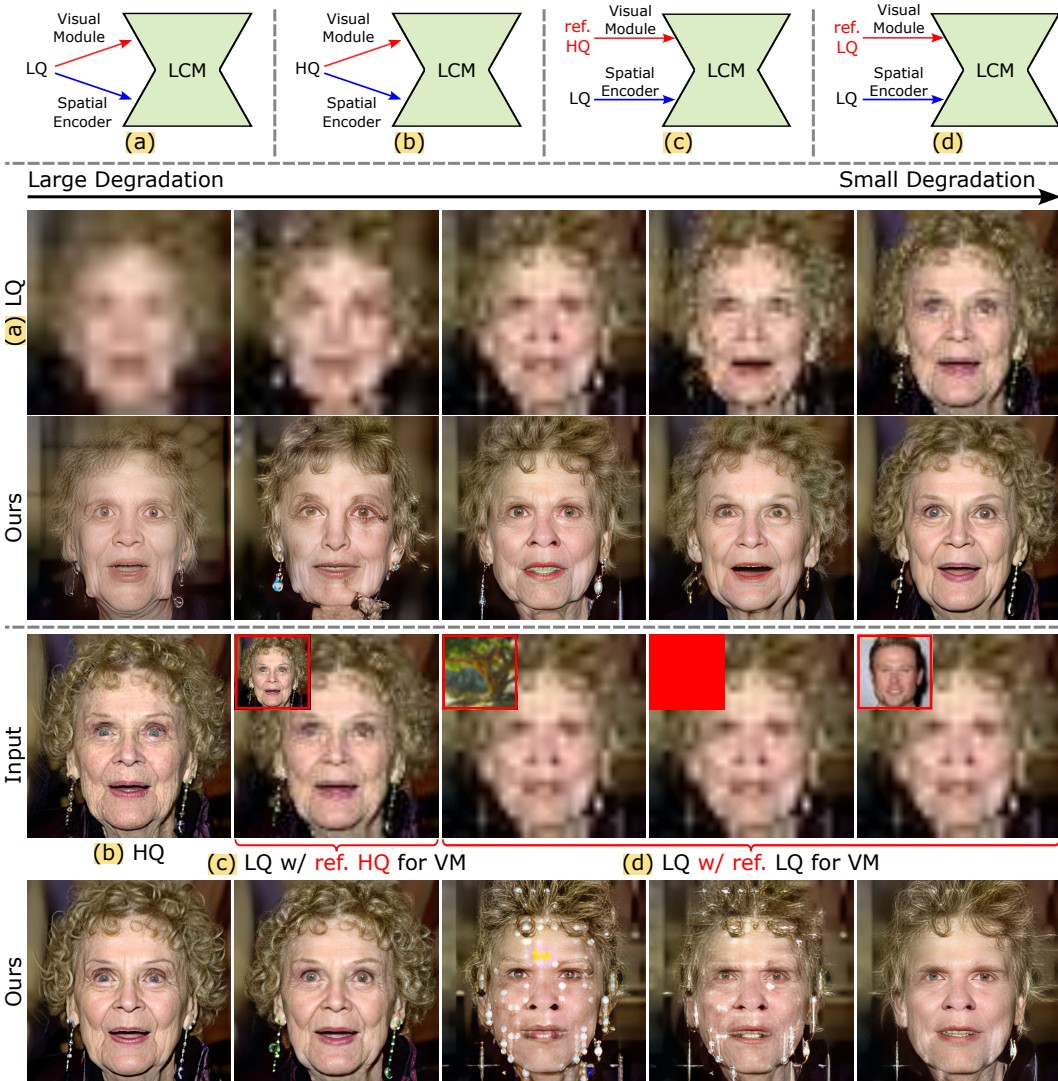

Figure 20: (top) Our method with variety inputs. (middle) We find that the semantic information from the LQ image suffices as a prior for HQ reconstruction when the degradation level is below a specific threshold (e.g., the third to fifth columns). (bottom) Using non-facial semantic images resulted in reconstructed outputs with artifacts (the third and fourth columns), whereas unrelated facial images provided sufficient semantic priors for generating HQ reconstructions with facial features (the fifth column).

indicates that the LQ image provides semantic information similar to that of the HQ image (Fig. 20, middle(the last column) vs., bottom(the first column)).

Than, we verify the provision of paired LQ and HQ images, which are provided to the Visual Module and Spatial Encoder (Fig. 20(c)). We also obverse that the reconstructed result shows similar semantic information to the HQ image (Fig. 20, bottom(the second column)).

To further assess the importance of facial semantic information from the LQ image for HQ reconstruction, we supplied the Visual Module with non-facial semantic images (Fig. 20, top(d)), such as non-facial semantic images (e.g., a image featuring a tree or a solid color) and unrelated facial images (Fig. 20, bottom(third and fifth columns)). Using non-facial semantic images resulted in reconstructed outputs with artifacts (Fig. 20, bottom(third and fourth columns)), whereas unrelated facial images provided sufficient semantic priors for generating HQ reconstructions with facial features (Fig. 20, bottom(fifth columns)).

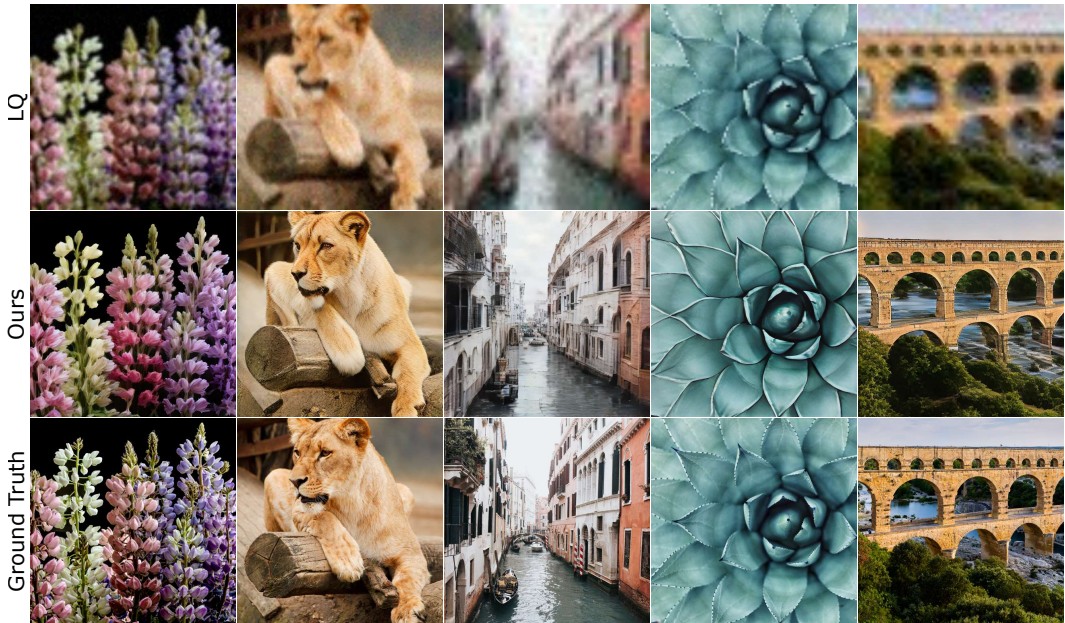

Figure 21: Results on natural image datasets.

## E.5 APPLYING THE PROPOSED METHOD TO NATURAL IMAGE DATASETS

For the blind face restoration problem, our method *InterLCM* can efficiently extract facial information through the Visual Encoder, as human faces are with less complex semantic information compared with real images from diverse scenarios. We show several real-image restoration results in Fig. 21. The results are satisfactory for simple textures, but less effective for complex textures. To improve the performance of our method *InterLCM* on real image, we plan to use a more powerful VQGAN-LC (Zhu et al., 2024) with 100,000 codebooks to act as the visual encoder for our model in future work.

## E.6 APPLYING PERCEPTUAL LOSS IN DIFFUSION-BASED MODELS

Several existing works (Chung et al., 2023; Laroche et al., 2024) have integrated the perceptual loss in to diffusion-based models. The forward process of diffusion-based models is a process that iteratively adds Gaussian noise to the representation using:

$$x_t = \sqrt{\alpha_t} x_{t-1} + \sqrt{1 - \alpha_t} \epsilon, \tag{4}$$

where $\alpha_t$ is the predefined variance, and $\epsilon \sim \mathcal{N}(\mathbf{0}, \mathbf{I})$. Recursively, let $\bar{\alpha}_t = \prod_{i=1}^{i=t} \alpha_i$, we have:

$$x_t = \sqrt{\bar{\alpha}_t} x_0 + \sqrt{1 - \bar{\alpha}_t} \epsilon. \tag{5}$$

When applying perceptual loss in diffusion-based models, the primary difference between our method and existing works (Chung et al., 2023; Laroche et al., 2024) lies in how the noise-free real image $x_0$ is obtained. Our approach uses $x_0$ at the final of the inference steps of the latent consistency model. In contrast, existing works (Chung et al., 2023; Laroche et al., 2024) derive $x_0$ from $x_t$ at an intermediate step $t$ by directly applying the inversion of forward process using Eq. (5):

$$\hat{x}_0 = \frac{1}{\sqrt{\bar{\alpha}_t}} (x_t - \sqrt{1 - \bar{\alpha}_t} \epsilon). \tag{6}$$

As shown in Fig. 22, we can observe that the $\hat{x}_0$ obtained from the SD intermediate steps (the first to fifth columns) has an appearance gap compared to the $x_0$ obtained using the full sampling process (the last column).

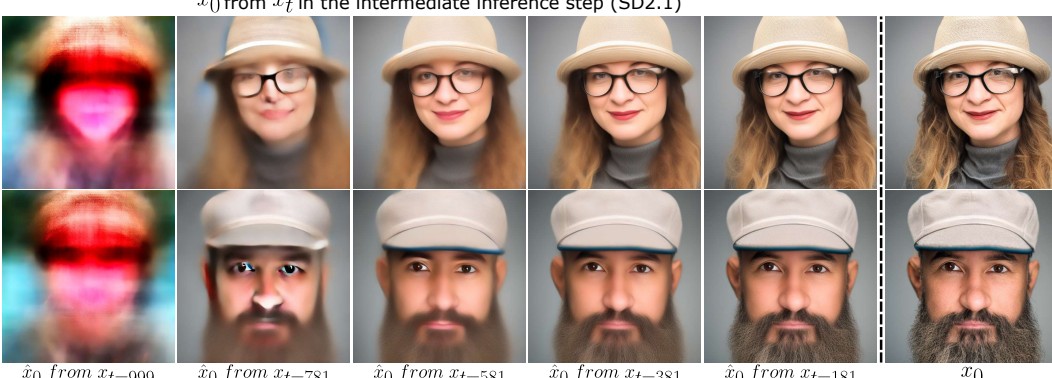

Figure 22: The $\hat{x}_0$ obtained from the intermediate step (the first to fifth columns) has an appearance gap compared to the $x_0$ (the last column).

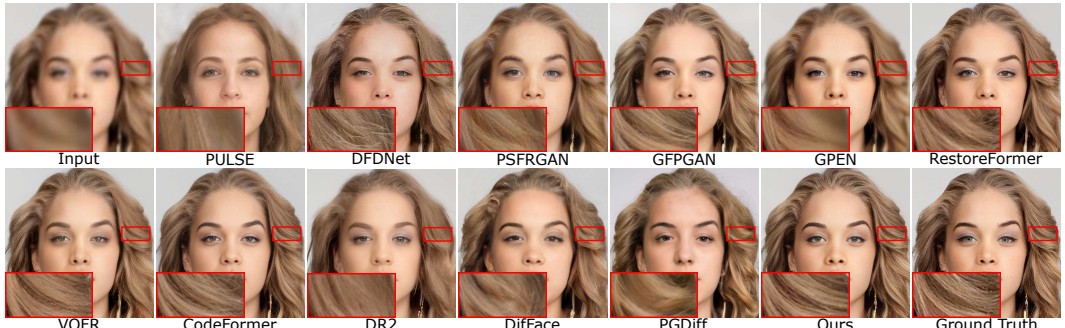

Figure 23: Qualitative comparisons of baselines on the synthetic of CelebA-Test for BFR.

## F ADDITIONAL RESULTS

As shown in Fig. 23, our method shows better hair quality than other methods and better aligns with the Ground Truth. Tab. 10 shows the quantitative comparison on the *synthetic* image of Fig. 23. Our method surpasses the baselines in two image quality metrics: MUSIQ and IDS. The Ground Truth has the best perceptual quality with the best MUSIQ metric 77.64. Actually, since the low-quality images are losing high-frequency information, the restoration is a random process to complement the high-frequency details (by varying seeds when adding noise).

We present additional qualitative comparisons of the baselines on real-world images from the LFW-Test, WebPhoto-Test, and WIDER-Test datasets in Fig. 24. As shown in Fig. 24, our method can reconstruct more realistic details in forehead wrinkles (first and second rows), eyes and eyebrows (third and fourth rows), and hair (fifth and sixth rows). These results demonstrate that our method outperforms the baselines in real-world scenarios.

In Figs. 25 to 28, we show additional reconstructed results on the synthetic dataset (i.e., CelebA-Test (Karras et al., 2017)) and the real-world dataset (i.e., LFW-Test (Huang et al., 2008), WebPhoto-Test (Wang et al., 2021a), and WIDER-Test (Yang et al., 2016)). Our *InterLCM* produces high-quality facial components and more realistic details compared to previous methods. We can generate high-quality images even under heavy degradation, while previous methods fail to do so (see Fig. 27 and Fig. 28).

Table 10: Quantitative comparison on the *synthetic* image of Fig. 23. The best results are in **bold**, and the second best results are underlined.

| Dataset | | *Synthetic* dataset Celeba-Test | | | |
|---|---|---|---|---|---|
| Method | Metrics | MUSIQ↑ | IDS↓ | PSNR↑ | SSIM↑ |
| Input | | 17.44 | 37.44 | 24.24 | 0.624 |
| CNN/Transformer-based | PULSE | 71.97 | 69.90 | 21.22 | 0.561 |
| | DFDNet | 75.96 | 27.42 | **25.03** | 0.620 |
| | PSFRGAN | 69.85 | 36.50 | 23.05 | 0.594 |
| | GFPGAN | 74.84 | 26.10 | 24.11 | 0.621 |
| | GPEN | 71.06 | 30.71 | 24.53 | **0.628** |
| | RestoreFormer | 75.57 | 26.52 | 23.69 | 0.595 |
| | VQFR | 74.23 | 32.97 | 23.70 | 0.598 |
| | CodeFormer | 76.19 | 28.55 | 24.25 | 0.612 |
| Diffusion-based | DR2 | 66.03 | 44.32 | 22.65 | 0.582 |
| | DifFace | 67.57 | 35.14 | 23.91 | 0.609 |
| | PGDiff | 69.44 | 54.98 | 22.35 | 0.586 |
| | **Ours** | **76.36** | **25.91** | 23.65 | 0.606 |

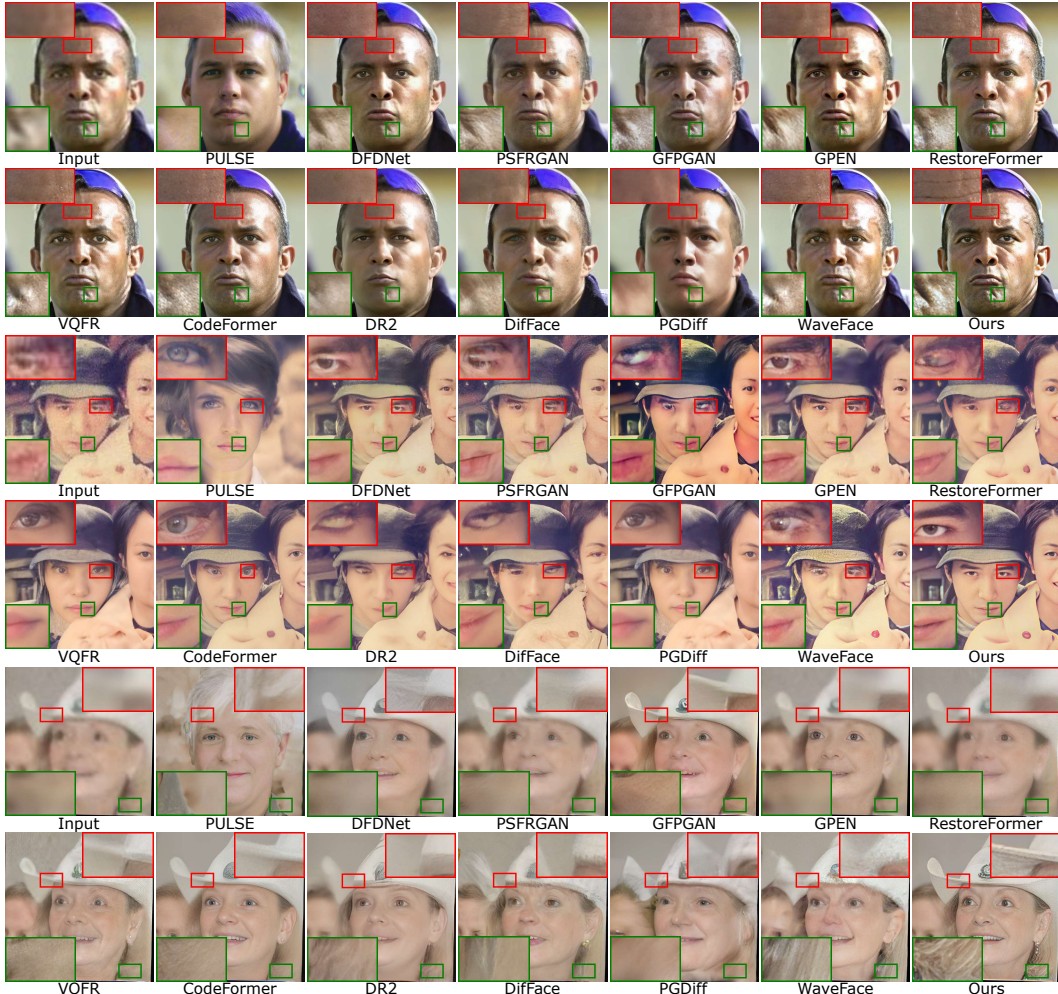

Figure 24: Qualitative comparisons of baselines on the real-world images from LFW-Test, WebPhoto-Test, and WIDER-Test. **Zoom in for a better view**.

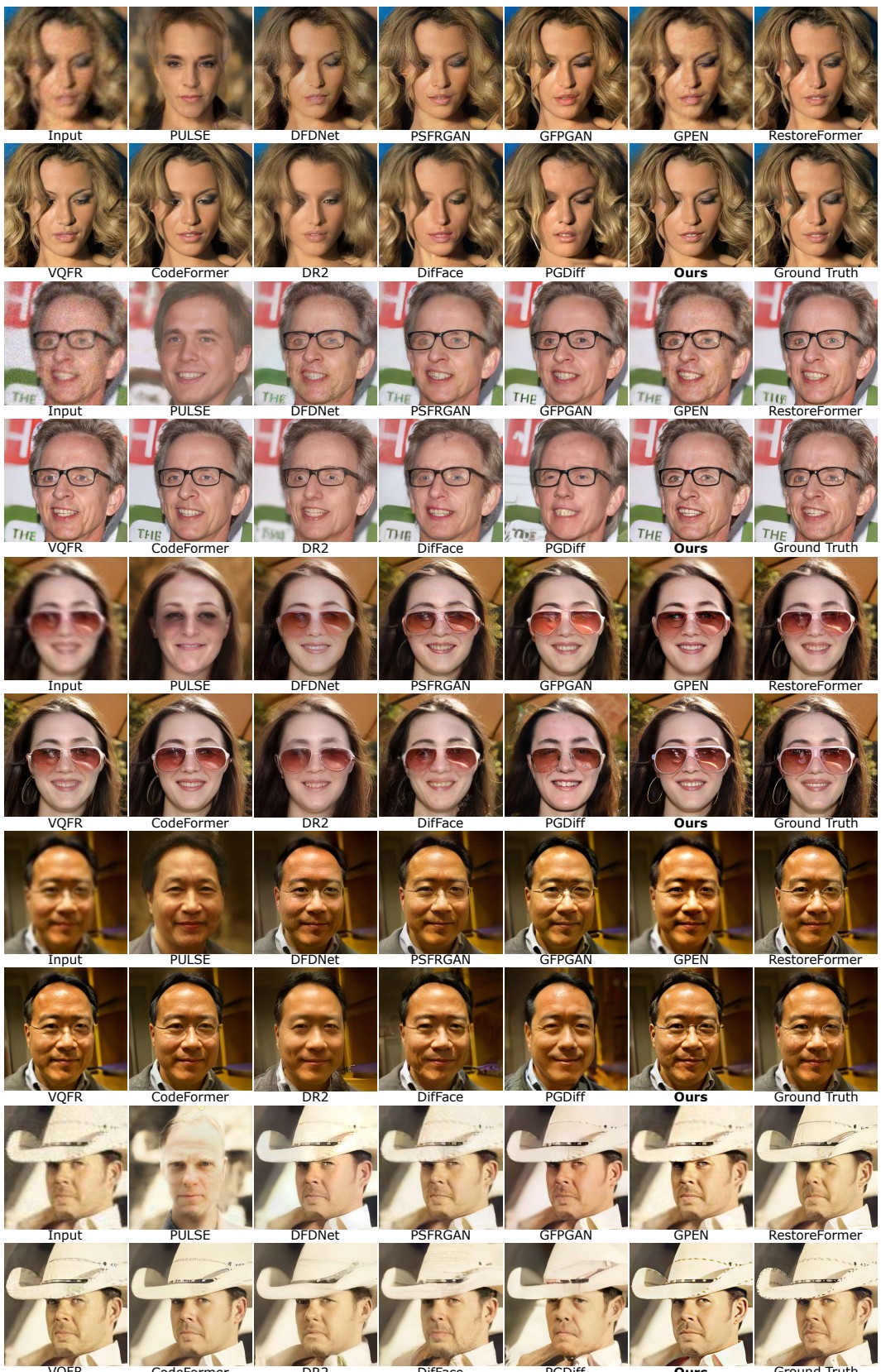

Figure 25: Qualitative comparison on the synthetic dataset Celeba-Test. **Zoom in for a better view**.

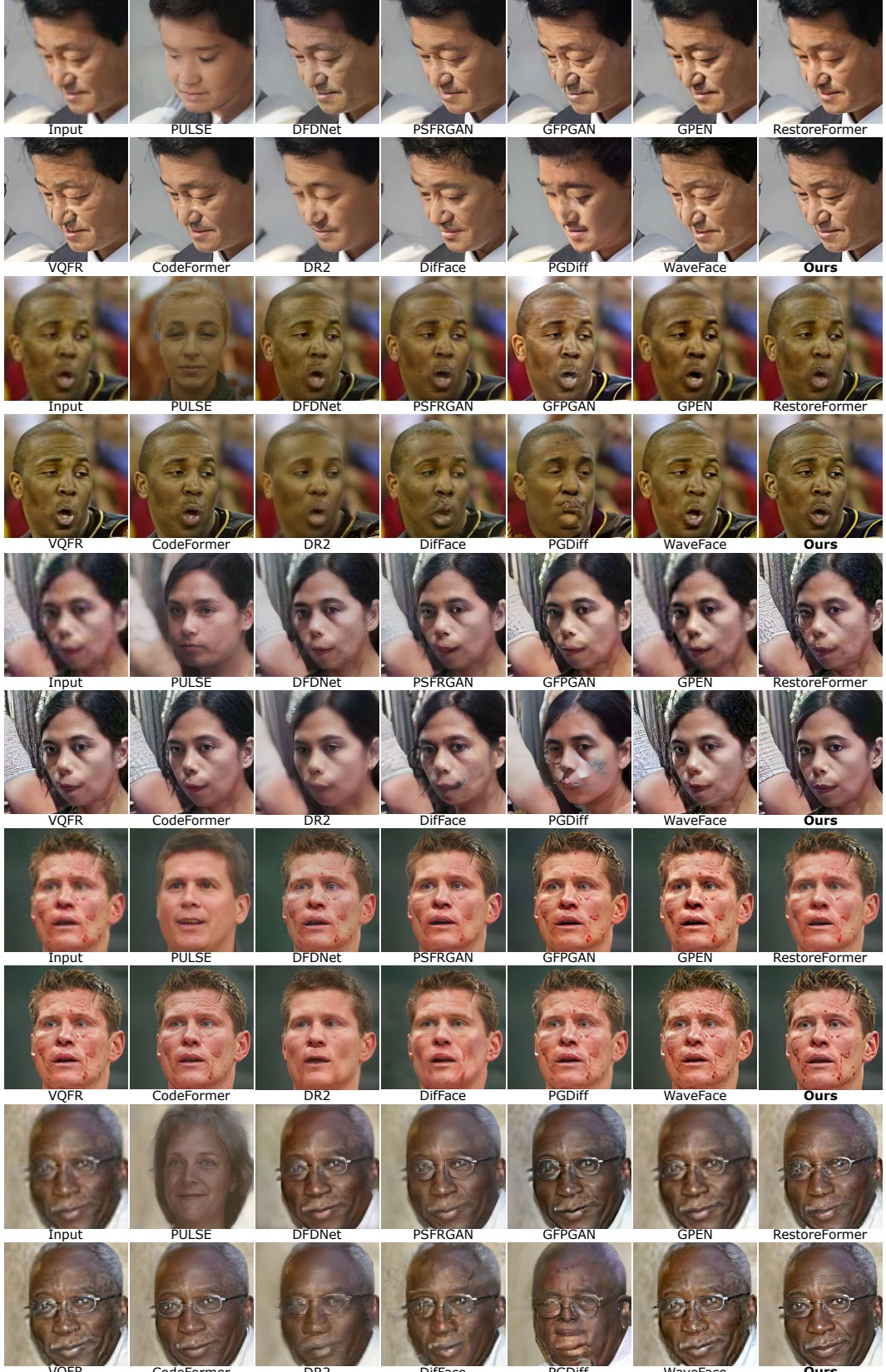

Figure 26: Qualitative comparison on the real-world dataset LFW-Test under mild degradation. **Zoom in for a better view**.

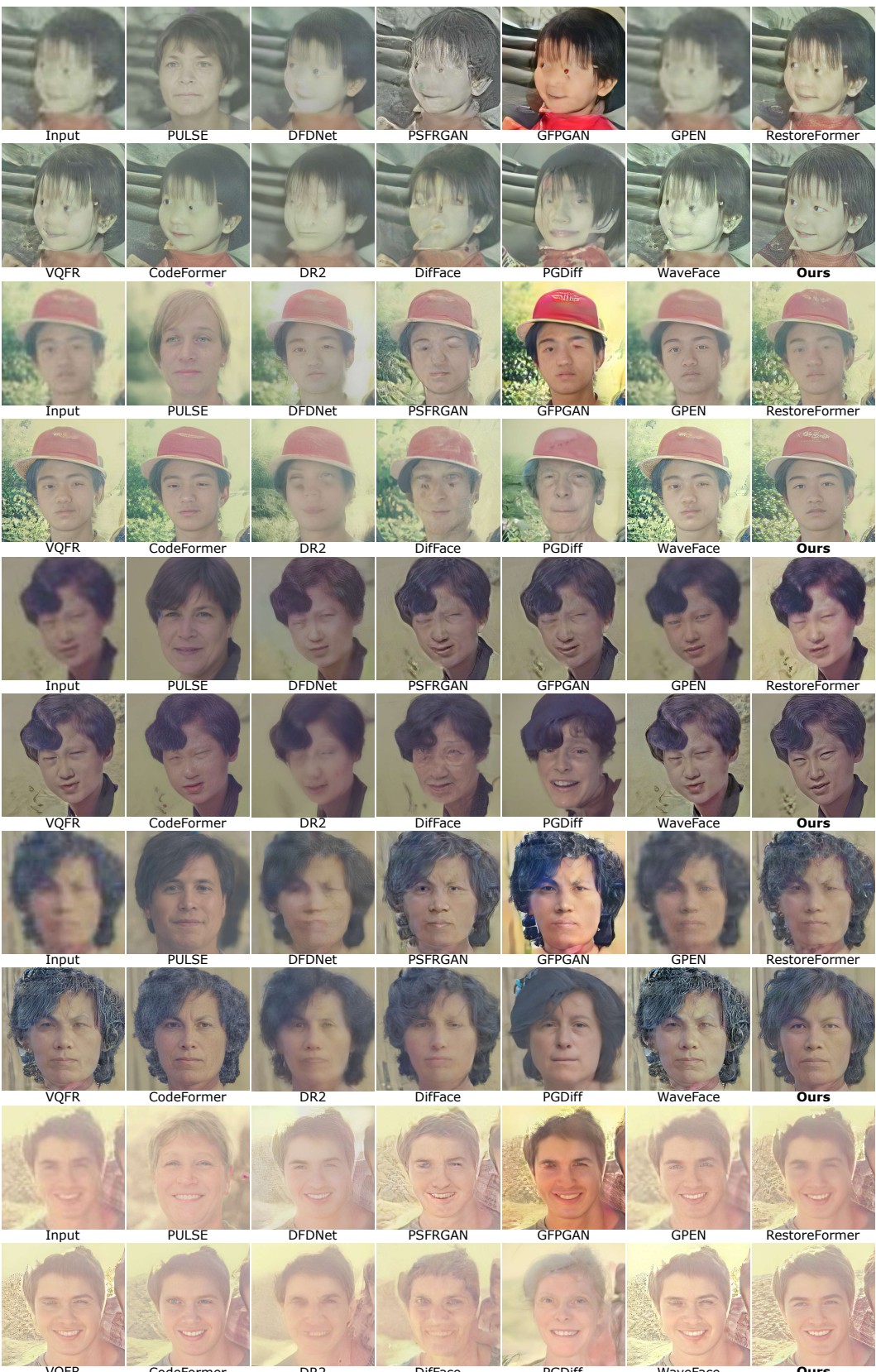

Figure 27: Qualitative comparison on the real-world dataset WebPhoto-Test under medium degradation. **Zoom in for a better view**.

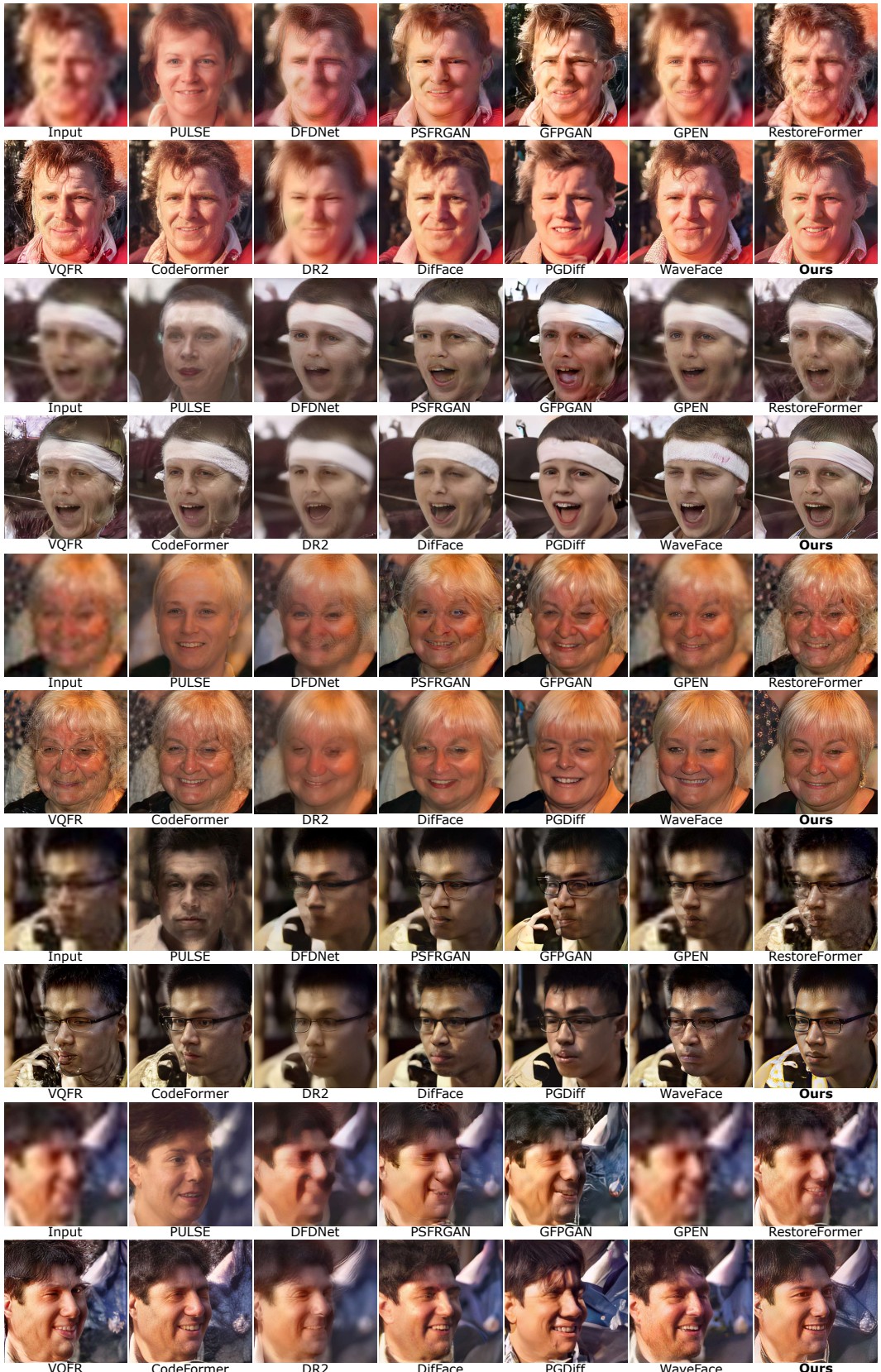

Figure 28: Qualitative comparison on the real-world dataset WIDER-Test under heavy degradation. **Zoom in for a better view**.

