# OpenReview forum: "$InterLCM$: Low-Quality Images as Intermediate States of Latent Consistency Models for Effective Blind Face Restoration"
_ICLR.cc/2025/Conference — ICLR 2025 Poster_

### Official Review · Reviewer_aK8y · 2024-10-28

**Soundness:** 3
**Presentation:** 3
**Contribution:** 3
**Rating:** 6
**Confidence:** 4

**Summary:**

The latent consistency model (LCM) demonstrates improved semantic consistency in subject identity, structural information, and color preservation, making it a viable alternative to conventional diffusion model priors. In this study, the authors propose InterLCM, which utilizes the low-quality image as an intermediate state in LCM models to facilitate efficient blind face restoration. Extensive experiments showcase the superior performance of InterLCM compared to existing approaches .

**Strengths:**

1.	the authors propose a LCM-based face restoration, a simple but effective BFR framework, it effectively maintain better semantic consistency in face restorations.
2.	LCM-based method shows faster inference than stable diffusion model

**Weaknesses:**

1.it is nice to find some trials on the LCM model for blind face restoration. But there exists some questions 1). Why prefer the bfr task not for the general sr task.2). the usage of lcm is straightforward, and the theory analysis and support is absent.
3).it is better to add more relevant works using the defined priors:

1). 3D Priors-Guided Diffusion for Blind Face Restoration
2). Scaling up to excellence: Practicing model scaling for photo-realistic image restoration in the wild
3). Face Restoration via Plug-and-Play 3D Facial Priors

**Questions:**

see in weakness

---

> ### Author Response · Authors · 2024-11-25
> **Response to Reviewer aK8y (1/1)**
>
> We appreciate your constructive comments and will incorporate the discussions mentioned below to enhance the quality of our paper.
>
> **W1(1): Why prefer the bfr task not for the general sr task?**
> For the blind face restoration problem, our method *InterLCM* can efficiently extract facial information through the Visual Encoder, as human faces are with less complex semantic information compared with real images from diverse scenarios.
> We show several real-image restoration results in **Figure.18**. The results are promising for simple textures, but less effective for complex textures. To improve the performance of our method on real image, we plan to use a more powerful VQGAN-LC with 100,000 codebooks to act as the visual encoder for our model in future work.
>
> **W1(2): The usage of lcm is straightforward, and the theory analysis and support is absent.**
> This work is inspired by observations from latent consistency models (LCMs), leveraging their priors alongside perceptual losses for blind face restoration. LCMs demonstrate superior semantic consistency, excelling in subject identity preservation, spatial structure retention, and color stability. In contrast, general diffusion models often fall short in these aspects, struggling with identity preservation, structural stability, and color consistency, as illustrated in **Figure.1**. By integrating image-level losses, such as perceptual loss and adversarial loss, our *InterLCM* model significantly improves blind face restoration (BFR) performance.
>
> **W1(3): It is better to add more relevant works using the defined priors**
> Thanks for your advice. We already include and discuss them in the introduction and related work of the updated version.

---

> > ### Comment · Reviewer_aK8y · 2024-11-26
> >
> > Thank you for the authors' feedback. These insights have addressed a significant portion of my concerns. As a blind face restoration technique, the introduction of novel priors has proven to be highly beneficial for advancing research in the field of face priors. However, certain concerns persist when extending these techniques to the broader domain of general super-resolution (SR).

---

> ### Author Response · Authors · 2024-11-28
> **New Response to Reviewer aK8y (1/1)**
>
> We appreciate your timely response and constructive comments. General super-resolution requires handling more complex semantic information, and we will explore general super-resolution tasks based on our framework in the future.

---

### Official Review · Reviewer_6W3D · 2024-10-31

**Soundness:** 2
**Presentation:** 2
**Contribution:** 2
**Rating:** 5
**Confidence:** 4

**Summary:**

The authors propose to use a latent consistency model to solve blind face restoration problems. Prior work on this research topic normally fine-tune a diffusion model on restoration datasets. To enhance the semantic consistency and incorporate the perceptual loss, authors further consider the low-quality image as the intermediate state of LCM models. The algorithm is tested on standard datasets and leads to improved performance over the considered baselines.

**Strengths:**

1. The paper is well-written, the experiments are systematic.
2. The authors manage to outperform the prior works on blind face restoration.

**Weaknesses:**

1. Regarding the contribution. Using pre-trained diffusion model as the image prior for blind face restoration is widely studied. The extension from conventional DDPM to LCM is quite straightforward.
2. Considering the combination of different training objectives, it is better to provide further analysis on the choices of hyper-parameters, and discuss about the risk of human bias due to these manual parameters.
3. I do not think the computation of perceptual loss is particularly challenging. Several existing works [1][2] on blind inverse problems have successfully integrated this term into a Bayesian framework.
[1] Parallel Diffusion Model of Operator and Image for Blind Inverse Problems (CVPR2023)
[2] Fast Diffusion EM: a diffusion model for blind inverse problems with application to deconvolution (WACV2024)

**Questions:**

1. Overall, I consider retraining to be a disadvantage. However, if the model demonstrates strong performance on a range of low-quality face images in real-world scenarios, it adds value. Can the authors make the trained weights available for testing?
2. How does the proposed method extend to images beyond just faces? If the study focuses solely on face restoration and is not even effective with images that include hands, I find it lacks generalizability and robustness.

---

> ### Author Response · Authors · 2024-11-25
> **Response to Reviewer 6W3D (1/2)**
>
> We appreciate your constructive comments and will incorporate the discussions mentioned below to enhance the quality of our paper.
>
> **W1: Using pre-trained diffusion model as the image prior for blind face restoration is widely studied. The extension from conventional DDPM to LCM is quite straightforward.**
> This work is inspired by observations from latent consistency models (LCMs), leveraging their priors alongside perceptual losses for blind face restoration. LCMs demonstrate superior semantic consistency, excelling in subject identity preservation, spatial structure retention, and color stability. In contrast, general diffusion models often fall short in these aspects, struggling with identity preservation, structural stability, and color consistency, as illustrated in **Figure.1**. By integrating image-level losses, such as perceptual loss and adversarial loss, our *InterLCM* model significantly improves blind face restoration (BFR) performance.
>
> **W2: The choices of hyper-parameters.**
> For the LCM inference step numbers, LCM commonly recommends to employ the 4-step inference process to balance image quality and inference time. In this paper, we use the recommended 4-step LCM, while we also offer an ablation study with a 2-step LCM model. As observed from the table below, the 4-step LCM only works slightly worse than the 2-step LCM on two metrics, which is utilized as the backbone for our *InterLCM*. For the hyperparameter \lambda, we set $\lambda=0.8$ following CodeFormer[A], where they also apply the perceptual and adversarial losses. This setup achieves a better trade-off for our BFR model in practice.
>  |         | Celeba-Test        |          |          |          |                |          | LFW-Test       |          | WebPhoto-Test |          |WIDER-Test|          |
> |---------|--------------------|----------|----------|----------|----------------|----------|----------------|----------|---------------|----------|----------|----------|
> |         | LPIPS↓             | FID↓     | MUSIQ↑   | IDS↓     | PSNR↑          | SSIM↑    | FID↓           | MUSIQ↑   | FID↓          | MUSIQ↑   | FID↓     | MUSIQ↑   |
> |Ours (2step LCM)| 0.248              | 49.19    | 75.31    | 34.92    | 23.91          | 0.662    | 56.21          |**76.24** | 75.84         | **76.11**| 38.23    | 76.00    |
> |Ours (4step LCM)| **0.223**          | **45.38**| **76.58**| **33.64**| **25.19**      | **0.718**| **51.32**      |76.16     | **75.48**     | 75.88    | **35.43**| **76.29**|
>
> [A] Towards Robust Blind Face Restoration with Codebook Lookup Transformer, NeurIPS 2022
>
> **W3: The computation of perceptual loss is particularly challenging. Several existing works [1][2] on blind inverse problems have successfully integrated this term into a Bayesian framework.**
> The challenge of the perceptual loss computation mainly lies in the accuracy and consistency of $x_0$ predicted in each inference step. Existing diffusion-based papers[1,2] derive $x_0$ from $x_t$ at an intermediate step by applying the inverse process of forward diffusion: ${\hat{x}\_0}=\frac{1}{\sqrt{\bar\alpha\_t}}(x\_t-\sqrt{1-\bar\alpha\_{t}}\epsilon)$.
>
> By comparison, when applying perceptual loss in diffusion-based models, our approach uses $x_0$​ at the end of the inference steps of the diffusion model, consistent with the standard diffusion model. In contrast, existing works [1,2] derive $x_0$ from $x_t$ at an intermediate step by applying the inverse process of forward diffusion. As shown in **Figure.19**, we can observe that the $x_0$​ obtained from the SD intermediate steps (the first to fifth columns) has an appearance gap compared to the $x_0$​ obtained using the full sampling process (the last column).
>
> [1] Parallel Diffusion Model of Operator and Image for Blind Inverse Problems (CVPR2023) \
> [2] Fast Diffusion EM: a diffusion model for blind inverse problems with application to deconvolution (WACV2024)

---

> > ### Author Response · Authors · 2024-11-25
> > **Response to Reviewer 6W3D (2/2)**
> >
> > **Q1: I consider retraining to be a disadvantage. However, if the model demonstrates strong performance on a range of low-quality face images in real-world scenarios. Can the authors make the trained weights available for testing?**
> >
> > We understand that retraining can be seen as a disadvantage, especially considering the computational resources required. We believe that our model demonstrates strong performance on a range of low-quality face images, particularly in real-world scenarios, as shown in **Table.1** (Real-world datasets) and **Figure.7** in the experiments section.
> >
> > Due to the anonymity policy, we are not easy to share an anonymous link to our pretrained weights during the reviewing process. To facilitate further testing and validation, we will make the code and trained weights publicly available.
> >
> > **Q2: How does the proposed method extend to images beyond just faces?**
> > For the blind face restoration problem, our method *InterLCM* can efficiently extract facial information through the Visual Encoder, as human faces are with less complex semantic information compared with real images from diverse scenarios.
> > We show several real-image restoration results in **Figure.18**. The results are promising for simple textures, but less effective for complex textures. To improve the performance of our method on real image, we plan to use a more powerful VQGAN-LC with 100,000 codebooks to act as the visual encoder for our model in future work.

---

> > > ### Comment · Reviewer_6W3D · 2024-11-28
> > >
> > > Thank you for the authors’ response, which has addressed some of my concerns. However, I still have remaining questions about the paper's novelty and believe that it might require further experimental validations and ablation studies.
> > >
> > > **W1**: As the authors also mentioned in their rebuttal, the use of perceptual loss and adversarial loss for training blind face restoration networks is not novel, as highlighted in [A]. I am not convinced that this can be treated as a major contribution of the paper.
> > > [A] Towards Robust Blind Face Restoration with Codebook Lookup Transformer, NeurIPS 2022
> > >
> > > **W2**: While the provided experiments are helpful, I believe more thorough ablation studies are necessary to isolate the contribution of each component. For example, how does the performance change if the perceptual loss is removed but the adversarial loss is retained?
> > >
> > > **Q1 & Q2**: My curiosity about the algorithm’s generalization to out-of-distribution (OOD) data stems from wanting to understand whether the retraining process is more akin to fine-tuning a pre-trained foundation model, potentially sacrificing generalizability by overfitting to a specific type of image, or if it genuinely learns a solver for general restoration tasks. Based on the current evidence, it seems to align more with the former, which leaves me uncertain about the broader impact of the algorithm.
> > >
> > > I’d be happy to hear more of the authors' thoughts on these questions, even qualitatively.

---

> > > > ### Author Response · Authors · 2024-11-30
> > > > **New Response to Reviewer 6W3D (1/2)**
> > > >
> > > > Thank you again for your follow-up response.
> > > >
> > > > **W1: As the authors also mentioned in their rebuttal, the use of perceptual loss and adversarial loss for training blind face restoration networks is not novel, as highlighted in [A]. I am not convinced that this can be treated as a major contribution of the paper. [A] Towards Robust Blind Face Restoration with Codebook Lookup Transformer, NeurIPS 2022**
> > > >
> > > > This work is mainly inspired by the observation that latent consistency models (LCMs) demonstrate greater semantic consistency in subject identity, spatial structure, and color preservation (see **Figure.1**). Based on the LCM property, we propose our method InterLCM, which regards the LQ image from the BFR problem as the input in an *intermediate step of LCM models* and obtains the high-quality image by performing the remaining few denoising steps (i.e., 3 steps) in 4-step LCM. To the best of our knowledge, we are the first to leverage the consistency properties (see **Figure 1**) of LCM for exploring BFR tasks, which we regard as the main contribution of this paper.
> > > >
> > > >
> > > > By this means, our method maintains better semantic consistency originated from the LCM. Meanwhile, benefitting from this property, we can integrate with both perceptual loss and adversarial loss, which are commonly used in restoration model training, leading to a high-quality and high-fidelity face restoration output. The use of perceptual loss and adversarial loss is our contribution based on the above observations of LCM properties.
> > > >
> > > > **W2: While the provided experiments are helpful, I believe more thorough ablation studies are necessary to isolate the contribution of each component. For example, how does the performance change if the perceptual loss is removed but the adversarial loss is retained?**
> > > >
> > > > We want to emphasize that this work is inspired by the observation that latent consistency models (LCMs) demonstrate greater semantic consistency in subject identity, spatial structure, and color preservation (see **Figure.1**). Meanwhile, benefitting from this property, we can integrate with both perceptual loss and adversarial loss, which are commonly used in restoration model training, leading to a high-quality and high-fidelity face restoration output.
> > > > We conducted an ablation study by removing the perceptual loss while retaining the adversarial loss. As shown in **the table below**, we can see that without perceptual loss (3), the quantitative metrics are significantly degraded, as it is challenging to reconstruct the textual detail using adversarial loss. This indicates that the perceptual loss plays a crucial role in the fidelity of the restored faces.
> > > >
> > > > Since it is not possible to update our submitted paper, we are now confirming the ICLR 2025 policy with Area Chairs to check the possibility to include more qualitative comparisons in anonymous links. Below, we include the quantitative comparisons in the table for your reference.
> > > >
> > > > | Exp. | $ℒ_1$ | $ℒ_{per}$ | $ℒ_{adv}$ | LFW-Test |   |WebPhoto-Test |   |WIDER-Test |   |
> > > > |------|--------------|-----------------|----------------|----------|--------|--------------|------------|-----------|------------|
> > > > |      |                 |                     |                   |FID↓      | MUSIQ↑ | FID↓         | MUSIQ↑     | FID↓      | MUSIQ↑     |
> > > > | (1) | ✓             |                    |                   |87.12    | 43.14   | 141.86       | 39.37      | 93.61     | 33.71      |
> > > > | (2) | ✓             | ✓                |                   |57.57    |  67.99  |95.02         | 66.24      | 44.83     | 63.94      |
> > > > | (3) | ✓             |                    |    ✓            |63.21    | 76.09   | 129.21       | 75.62      | 100.15    | 74.40      |
> > > > | (4) Ours  | ✓     | ✓               | ✓               |**51.32** |**76.16**|**75.48** |**75.88**|**35.43** |**76.29** |

---

> > > > > ### Author Response · Authors · 2024-11-30
> > > > > **New Response to Reviewer 6W3D (2/2)**
> > > > >
> > > > > **Q1 & Q2: My curiosity about the algorithm’s generalization to out-of-distribution (OOD) data stems from wanting to understand whether the retraining process is more akin to fine-tuning a pre-trained foundation model, potentially sacrificing generalizability by overfitting to a specific type of image, or if it genuinely learns a solver for general restoration tasks. Based on the current evidence, it seems to align more with the former, which leaves me uncertain about the broader impact of the algorithm.**
> > > > >
> > > > > As shown in **Figure 1** and **Figure 5**, the large pretrained T2I models themselves already have the capability to restore the LQ image by their pretrained knowledge from billions of image-text pairs. In our case, we are refining their restoration quality by finetuning extra visual encoder (VE) and spatial encoder (SE) over the BFR datasets FFHQ with 70,000 images, which are comparably much smaller than their pretrained datasets. In this procedure, the LCM backbone is kept frozen. By this method proposed in our paper, InterLCM can efficiently address the blind face restoration (BFR) problem through the optimized VE and SE modules. More detailed descriptions of the our training and evaluation are included below.
> > > > > Furthermore, we demonstrate the possibility to generalized our method InterLCM to general image restoration tasks in **Figure.20** and aim to improve the performance with much stronger VQGAN-LC architectures for wider applications. The generalizability of LCM is not sacrificed, instead it can be further explored in our future work.
> > > > >
> > > > > **Training details.** We also want to emphasize that this work is inspired by the observation that latent consistency models (LCMs) demonstrate greater semantic consistency in subject identity, spatial structure, and color preservation (see **Figure.1**). In the training phase, the pretrained LCM model is frozen to maintain the consistency properties, while the Visual Encoder and Spatial Encoder are trainable. First, a CLIP image encoder and Visual Encoder as Visual Module that helps to extract semantic information from faces, providing the LCM with face-specific priors. Second, to prevent changes in content (e.g., structure), we include a Spatial Encoder to leverage the strong semantic consistency of the LCM model.
> > > > >
> > > > > **Evaluation details.** Based on a data-driven approach, we use the commonly used FFHQ dataset in blind face restoration (BFR) methods (e.g., [A]) and achieve state-of-the-art (SOTA) results across multiple metrics. Both qualitative and quantitative experimental results demonstrate that our method performs better on both synthetic and real datasets. Our method also faces challenges when applied to images that include hands. It excels in generating detailed facial features but does not produce realistic hands (see **Figure 10**). We demonstrate the ability to reconstruct high-quality details even in challenging cases, such as images featuring tattoos or festival-style face paint (see **Figure.18**).
> > > > >
> > > > > [A] Towards Robust Blind Face Restoration with Codebook Lookup Transformer, NeurIPS 2022

---

> > > > > > ### Author Response · Authors · 2024-12-02
> > > > > >
> > > > > > Dear Reviewer **6W3D**,
> > > > > >
> > > > > > Thank you once again for your constructive comments. Your constructive comments have been very helpful.
> > > > > >
> > > > > > In response to your remaining questions, we have clarified the major contributions and provided more implementation details of the paper. Additionally, we conducted an additional ablation study. We would like to know if our responses have addressed your concerns.
> > > > > >
> > > > > > **As the rebuttal window is closing soon, we genuinely appreciate your feedback. We hope our response has addressed your concerns.**
> > > > > >
> > > > > > Best Regards,
> > > > > > Authors of submission 536

---

> > > > > > ### Comment · Reviewer_6W3D · 2024-12-02
> > > > > >
> > > > > > Thank you for your detailed rebuttal and the kind reminder. I appreciate the additional experimental results. However, after reviewing your response, my concerns about the contributions and generalizability remain unresolved.
> > > > > >
> > > > > > **Contribution**: The work appears to combine existing techniques, including LCM, perceptual loss, and adversarial loss, to address a relatively straightforward problem on the face-only dataset.
> > > > > >
> > > > > > **Generalizability**: The observations in Figure 20 do not exhibit significant corruption, which limits the demonstration of the algorithm's generalizability. Even when considering face images, performance tends to degrade with images involving hands. Additionally, I am not requesting detailed training or evaluation information, so there's no need to include these specifics.
> > > > > >
> > > > > > Considering the above concerns, I decide to maintain my current evaluation.

---

> > > > > > > ### Author Response · Authors · 2024-12-03
> > > > > > > **New Response to Reviewer 6W3D (1/1)**
> > > > > > >
> > > > > > > Thank you for your response. We will carefully consider your suggestions, which have been very helpful for our paper.
> > > > > > >
> > > > > > > **Contribution**:
> > > > > > > We would like to emphasize that our method is not merely a combination of existing techniques. Instead, we leverage the semantic consistency of LCM models and utilize the inherent noise injection of LCM to invert low-quality (LQ) images into the noise space. Our method using LCM effectively addresses the distribution gap issue present in other methods when inverting to the noise space, thereby making our method more suitable for combining perceptual loss and adversarial loss.
> > > > > > >
> > > > > > > **Generalizability**:
> > > > > > > Our method builds upon the discovery of the semantic continuity property of LCM models. Similar to state-of-the-art (SOTA) methods like CodeFormer, DifFace, and WaveFace, which primarily focus on facial reconstruction tasks. The failure to reconstruct hands in **Figure.10** is due to the ambiguous semantics of hands in the LQ images. These SOTA methods also fail to reconstruct such cases successfully.

---

### Official Review · Reviewer_Hm1E · 2024-11-02

**Soundness:** 3
**Presentation:** 4
**Contribution:** 3
**Rating:** 6
**Confidence:** 4

**Summary:**

This paper proposes to regard the low-quality image as the intermediate state of LCM models for blind face restoration, considering that the LCM enjoy superior semantic consistency compared to the naive diffusion models, in terms of the generative prior utilization. Experiments demonstrate the effectiveness of the proposed method.

**Strengths:**

1. This paper proposes an efficient method to utilize the generative prior for blind face restoration, that the LCM models enjoy more semantic consistency compared to the naive diffusion models, and the image-space optimization benefit more restoration-specific loss constrains.
2. The semantic consistency comparison between LCM and diffusion models are well illustrated, and the method design is straightforward without bells and whistles. The intermediate state validation and ablation experiments are reasonable and sufficient.
3. The presentation is well for readability.

**Weaknesses:**

1. Is there any comparison for x0-prediction-based diffusion model.
2. Why only face dataset, whether the proposed method could generalize to other natural image dataset, is there any discrepancy in adopting the LCM models as generative prior. If so, what modifications should we care to apply current method to other types of images, and the preliminary results on a non-face dataset would be nice if feasible.
3. Whether the proposed method can be integrated with LCM-LoRA for more fast inference? Is there any potential challenge, and the preliminary results would be recommended if feasible.

**Questions:**

1. The training efficiency is wondered, as the LCM backbone is freezed, how long could we get the model, and the training stability is concerned when various loss functions are added.
2. What's the difference between the spatial encoder and ControlNet, only incorporating the visual embedding?

---

> ### Author Response · Authors · 2024-11-25
> **Response to Reviewer Hm1E (1/1)**
>
> We appreciate your constructive comments and will incorporate the discussions mentioned below to enhance the quality of our paper.
>
> **W1: Is there any comparison for x0-prediction-based diffusion model.**
> To the best of our knowledge, the Consistency Model (CM) is the only diffusion model that features a x0-prediction branch. Unlike DDPM models, which incrementally perform denoising across multiple steps, CM directly learns to map $z_t$ to the original real image space $z_0$.
> Taking SD-Turbo as a representative method in xt-prediction-based model, we test the SD-Turbo as the backbone to develop our method for the BFR problem. By the quantitative comparison in **Table.6** in the updated Appendix, we show that consistency model, which directly predict the $x_0$ in each step, better suits the BFR problem.
>
> **W2: Whether the proposed method could generalize to other natural image dataset?**
> For the blind face restoration problem, our method *InterLCM* can efficiently extract facial information through the Visual Encoder, as human faces are with less complex semantic information compared with real images from diverse scenarios.
> We show several real-image restoration results in **Figure.18**. The results are promising for simple textures, but less effective for complex textures. To improve the performance of our method on real image, we plan to use a more powerful VQGAN-LC with 100,000 codebooks to act as the visual encoder for our model in future work.
>
> **W3: Whether the proposed method can be integrated with LCM-LoRA for more fast inference?**
> The distillation process of LCM updates the entire model's parameters. In contrast, the distillation process of LCM-LoRA is performed only on the parameters of two low-rank matrices, while the pre-trained diffusion model remains frozen.
> The advantage of LoRA is to reduce the memory and compute overhead during training. After training, the LoRA parameters are added up to the pre-trained diffusion model before inference. During inference, LCM and LCM-LoRA have the similar number of parameters, resulting in close memory usage (6332.54MB vs. 6334.26MB) and inference time on a single 3090 GPU. Therefore, when combined with our *InterLCM*, either using the frozen LCM or LCM-LoRA, the memory usage (8334.56.MB vs. 8599.13MB) and inference time remain comparable.
>
> **Q1: How long could we get the model?**
> The *InterLCM* models are trained for 15K iterations using eight A40 GPUs (48GB VRAM), which takes nearly 9.46 hours for the saturation of the loss convergence.
>
> **Q2: What's the difference between the spatial encoder and ControlNet, only incorporating the visual embedding?**
> Our Spatial Encoder shares a similar structure and feature injection mechanism with the original ControlNet but differs in two key aspects:
>
> **(a)** *Integration of Image-Level Losses:* LCMs enable the integration of image-level losses, such as perceptual loss and adversarial loss. In contrast, using ControlNet with the original diffusion loss (Eq. 1) proves ineffective for face restoration, as shown in **Table.4**. By leveraging the ControlNet architecture—replicated from the Stable Diffusion encoder modules and serving as the Spatial Encoder (SE) in our method, *InterLCM*—alongside the perceptual loss (Eq. 2), we achieve significant improvements in blind face restoration (BFR) performance.
>
> **(b)** *Input Type:* The original ControlNet primarily takes canny edges or depth maps as input. Naively applying ControlNet for blind face restoration, as shown in **Figure.9** (the fifth to seventh columns), involves using low-quality (LQ) images as input. While this approach generates high-quality outputs that preserve structural integrity, it often compromises fidelity.

---

> ### Comment · Reviewer_Hm1E · 2024-11-27
> **Official Comment by Reviewer Hm1E**
>
> Thanks for the authors' response, which has address part of my concerns. And there remain some concerns that unresolved.
>
> Basically, the x0-prediction and consistency models are different model family with different noise schedules in training and different one-step generation ability. It may better to include the discussion with x0-prediction models, and show some comparisons.
>
> LCM-LoRA is integrated for accelerated inference time with reduced sampling steps, and the memory usage the author presented are somewhat less concerned.  I wonder whether the integration of the LCM-LoRA works for the proposed method, and the visual comparison or other quantitative comparison would be better.

---

> ### Author Response · Authors · 2024-11-28
> **New Response to Reviewer Hm1E (1/1)**
>
> Thank you for your response and the insightful question.
>
> **1. The x0-prediction and consistency models are different model family with different noise schedules in training and different one-step generation ability. It may better to include the discussion with x0-prediction models, and show some comparisons.**
>
> We use one-step models ($x_0$-prediction-based diffusion models) as the backbone to develop our method for the BFR problem. We first move the LQ image to the noise space of the one-step models. We make some comparisons in Table below (**Table.8** in main paper) and **Figure.14**. As shown in Table below, our metrics significantly outperform one-step diffusion models in the BFR task, except for the FID metric on the Synthetic dataset. As shown in the qualitative comparison in **Figure.14**,  results of our method using one-step models (as shown in the second and third rows) indicate that these models face challenges with artifacts and blur when reconstructing high-quality images, while our method can reconstruct high-quality images with detailed textures (the fourth row).
>
> | Dataset   |Synthetic dataset|  |   |  |    |    |Real-world datasets |   |   |  |    |   |
> |-----------|-----------------|--|---|--|----|----|---|--|----|---|----|---|
> | Metrics   | Celeba-Test     |  |   |  |    |    | LFW-Test |  | WebPhoto-Test|  | WIDER-Test|    |
> | Method    | LPIPS↓          | FID↓    | MUSIQ↑     | IDS↑       | PSNR↑      | SSIM↑      | FID↓     |  MUSIQ↑  |   FID↓  |  MUSIQ↑  |   FID↓  | MUSIQ↑  |
> | Input                  | 0.574             | 145.22  | 72.81      | 47.94      | 22.72      | 0.706      | 138.87      | 26.87      | 171.63       | 18.63 | 201.31    | 14.22  |
> | Ours (1-step SD Turbo)                 | 0.273             | **36.87**   | 74.00      | 37.82      | 24.89       | 0.659     | 61.21       | 70.24      | 87.77        | 70.47 | 54.45     | 71.51  |
> | Ours (1-step LCM)      |     0.240   |  46.66 | 74.06   |  37.45    |   24.66    |   0.697  |    55.7  |   73.45   |   89.90     |   72.41  |     37.16 | 70.45|
> | Ours (4-step LCM)             | **0.223**         | 45.38   | **76.58**  | **33.64**  | **25.19**     | **0.718** | **51.32**       |**76.16**  | **75.48**    |**75.8**|**35.43**|**76.29**|
>
> **2. LCM-LoRA is integrated for accelerated inference time with reduced sampling steps, and the memory usage the author presented are somewhat less concerned. I wonder whether the integration of the LCM-LoRA works for the proposed method, and the visual comparison or other quantitative comparison would be better.**
>
> We test the LCM-LoRA as the backbone to develop our method for the BFR problem. Table below (**Table.7** in main paper) and **Figure.13** show the qualitative and quantitative, respectively, comparison of using LCM-LoRA and our method. LCM-LoRA does not perform as well as our method in terms of LPIPS and FID metrics, while it achieves better results on the MUSIQ metric for image quality evaluation on real datasets, such as LFW-Test and WebPhoto-Test.  The qualitative results in **Figure.13** demonstrate that both LCM-LoRA and our method can achieve high-quality reconstructed images.
>
> | Dataset   |Synthetic dataset|  |   |  |    |    |Real-world datasets |   |   |  |    |   |
> |-----------|-----------------|--|---|--|----|----|---|--|----|---|----|---|
> | Metrics   | Celeba-Test     |  |   |  |    |    | LFW-Test |  | WebPhoto-Test|  | WIDER-Test|    |
> | Method    | LPIPS↓          | FID↓    | MUSIQ↑     | IDS↑       | PSNR↑      | SSIM↑      | FID↓     |  MUSIQ↑  |   FID↓  |  MUSIQ↑  |   FID↓  | MUSIQ↑  |
> | Input                  | 0.574             | 145.22  | 72.81      | 47.94      | 22.72      | 0.706      | 138.87      | 26.87      | 171.63       | 18.63 | 201.31    | 14.22  |
> | Ours (LCM-LoRA)                 | 0.240             | 53.26   | **76.58**      | 35.48      | 24.14       | 0.661     | 54.70       | **76.26**      | 82.08        | **76.59** | 39.62     | 75.81  |
> | Ours (LCM)             | **0.223**         | 45.38   | **76.58**  | **33.64**  | **25.19**     | **0.718** | **51.32**       | 76.16  | **75.48**    |75.8|**35.43**|**76.29**|

---

> > ### Author Response · Authors · 2024-12-02
> >
> > Dear Reviewer **Hm1E**,
> >
> > Thank you for dedicating your time to review our paper and your valuable suggestions. We are pleased to address all your concerns in the rebuttal period.
> >
> > **As the rebuttal window is closing soon, we genuinely appreciate your feedback. We hope our response has addressed your concerns.**
> >
> > Best Regards,
> > Authors of submission 536

---

### Official Review · Reviewer_43VF · 2024-11-04

**Soundness:** 2
**Presentation:** 3
**Contribution:** 2
**Rating:** 3
**Confidence:** 4

**Summary:**

This paper proposes a new method for blind face image restoration, consisting of three main components: 1) applying a latent consistency model (LCM) for fast sampling; 2) starting from a low-resolution (LR) image instead of standard Gaussian noise during inference; and 3) incorporating semantic information from the LR image. The authors compare their method against numerous baselines and achieve better quantitative performance over most metrics.

**Strengths:**

1) The comparison against various baselines is comprehensive and well-organized.
2) Figure 2 and Figure 3 clearly illustrate the key concepts of the method.
3) The qualitative results demonstrate a significant improvement over several baselines.

**Weaknesses:**

1) The paper overlooks important literature [1], which discusses accelerating diffusion models for inverse problems, including image restoration, by beginning with a better initialization rather than standard Gaussian noise. This prior work shares a similar concept of initiating the diffusion process from LR images in this study.
2) One motivation for employing LCM appears to be the acceleration of the diffusion prior. However, the manuscript simply states that “diffusion-prior based approaches still suffer from time-consuming inferences” (line 161) without discussing the extensive literature on accelerating diffusion models, such as techniques involving distillation (beyond the consistency model) and diffusion bridges [2].
3) The abstract is difficult to follow. I recommend revising it to make it shorter and more focused, allowing readers to easily grasp the key ideas of the paper.
4) While the quantitative metrics indicate improvements, the visual results do not convincingly demonstrate superior performance. For instance, compared to the ground truth, the proposed method generates features that differ significantly (e.g., hair in Figure 6), similar to other baselines. From a perceptual quality standpoint, it is challenging to determine whether the proposed method is indeed better. In Figure 6, the proposed method oversharpens the image compared to the reference. This concern is further supported by only marginal improvements in perceptual metrics over the baseline methods, such as CodeFormer (with a worse FID score than some baselines and a marginal improvement of 0.004 in LPIPS and 1.03 in MUSIQ).
5) Some statements in the manuscript appear overly assertive without adequate support from references or experimental evidence. For example, the abstract claims that “the latent consistency model shows more semantic consistency in the subject identity,” and line 163 states that “the commonly used perceptual loss in image restoration tasks cannot be well integrated into their framework.”
6) The contributions of this paper seem somewhat incremental. The use of LR as a prior has been investigated in [1], and LCM for fast diffusion prior is a well-established technique. From this perspective, the main contribution appears to be the incorporation of semantic information from LR images, which raises questions about whether it meets the standards for ICLR.

[1] Chung, Hyungjin, Byeongsu Sim, and Jong Chul Ye. "Come-closer-diffuse-faster: Accelerating conditional diffusion models for inverse problems through stochastic contraction." Proceedings of the IEEE/CVF Conference on Computer Vision and Pattern Recognition, 2022.

[2] Liu, Guan-Horng, et al. "I $^ 2$ SB: Image-to-Image Schrödinger Bridge." International Conference on Machine Learning. PMLR, 2023.

**Questions:**

1) Could the authors provide examples of the semantic information from HR and LR images of the same scene such that one could evaluate whether the semantic information from the LR image is sufficient as a prior for HR reconstruction?
2) How were the hyperparameters (2) chosen in this study?
3) Why was a four-step LCM considered instead of a different number of steps? This choice seems heuristic.
4) Figure 1 is unclear. It says that LCM maps directly to the real image space. Does it imply LCM learns a mapping from LR to HR images directly like [2], rather than progressively denoising as depicted in Figures 2 and 3?
5) What is the degradation process considered in the synthetic dataset? Is it a simple interpolation?

---

> ### Author Response · Authors · 2024-11-25
> **Response to Reviewer 43VF (1/3)**
>
> We appreciate your constructive comments and will incorporate the discussions mentioned below to enhance the quality of our paper.
>
> **W1&W2&W6: Discusses accelerating and the use of LQ as a prior.**
> First, using the intermediate state as a better initialization for diffusion tasks is a common practice in existing works, such as SDEdit[A], DR2[B], and DifFace[C]. However, these approaches often suffer from issues like distribution gaps[D] and signal leakage[E]. In contrast, the LCM model directly maps the noise space to the real image $z_0$, effectively bridging the distribution gap.
>
> As to your referred papers, the literature [1,2] aims to accelerate diffusion-based models. In contrast, our method, *InterLCM*, is inspired by the latent consistency model (LCM), which demonstrates superior semantic consistency, including subject identity preservation, spatial structure retention, and color stability. However, general diffusion models, including distilled models, often struggle with maintaining semantic consistency, such as identity consistency, structural stability, and color preservation (see **Figure.1**).
>
> Moreover, our few-step sampling approach enables *InterLCM* to integrate perceptual loss and adversarial loss for face restoration tasks. While speed improvement is an inherent benefit of the LCM model's properties, it is not our primary contribution. Additionally, we use low-quality (LQ) images as inputs to our proposed Visual Encoder (VE) and Spatial Encoder (SE), which extract semantic and visual information as conditions to enhance face restoration, rather than merely using LQ images as training inputs.
>
> Finally, blind face restoration (BFR) is distinct from other restoration tasks like super-resolution, deblurring, and JPEG restoration, as discussed in [1,2]. BFR addresses complex, real-world degradations (see Appendix A.2), whereas other tasks typically deal with more straightforward degradations.
>
> We already put the discussion about [1,2] in the related work of the updated submission.
>
> [A] SDEdit: Guided Image Synthesis and Editing with Stochastic Differential Equations. ICLR2022 \
> [B] DR2: Diffusion-based Robust Degradation Remover for Blind Face Restoration. CVPR2023 \
> [C] DifFace: Blind Face Restoration with Diffused Error Contraction. TPAMI, 2024 \
> [D] Exploiting the signal-leak bias in diffusion models. WACV2024 \
> [E] Real-world image variation by aligning diffusion inversion chain. NeurIPS2023 \
> [1] Come-closer-diffuse-faster: Accelerating conditional diffusion models for inverse problems through stochastic contraction. CVPR2022 \
> [2] I2SB: Image-to-Image Schrödinger Bridge. ICML2023
>
> **W3: The abstract is difficult to follow.**
> Thanks for your suggestion. We have modified the abstraction in the updated version.
>
> **W4: While the quantitative metrics indicate improvements, the visual results do not convincingly demonstrate superior performance.**
> The blind face restoration is based on the whole dataset statistics, not merely focusing on improving the restoration quality of a single person. As shown in **Table.1**, for the evaluation of the synthetic and the real-world datasets, our method achieves the best performance on two metrics IDS and  MUSIQ for identity and quality metric.
>
> **In Figure.6**, our method significantly outperforms others in capturing textural details, despite differing from the Ground Truth. The reason for this difference is that the Ground Truth in this case contains natural noise, while restoration methods based on generative priors struggle to produce results consistent with the Ground Truth. We include more examples in Appendix with detailed metrics to show that our method outperforms other approaches.
>
> As shown in **Figure.20**, our method shows better hair quality than other methods and better aligns with the Ground Truth. Actually, since the low-quality images are losing high-frequency information, the restoration is a random process to complement the high-frequency details (by varying seeds when adding noise).

---

> > ### Author Response · Authors · 2024-11-25
> > **Response to Reviewer 43VF (2/3)**
> >
> > **W5: Some statements seem overly assertive without sufficient support. For instance, the abstract claims “the latent consistency model shows more semantic consistency in the subject identity,” and line 163 states “the commonly used perceptual loss in image restoration tasks cannot be well integrated into their framework.”**
> > The observation that the latent consistency model demonstrates greater semantic consistency is presented in **Figure.1**. Specifically, **Figure.1** (right) evaluates identity consistency across adjacent steps. Integrating perceptual loss into diffusion models poses challenges because these models are designed to predict the noise and the preceding latent state $z_{t-1}$​ from $z_t$, rather than the real image at each step. In contrast, consistency models are trained to directly predict $z_0$ from $z_t$ at each step, making the incorporation of perceptual loss more straightforward.
> >
> > **Q1: Could the authors provide examples of the semantic information from HR and LR images of the same scene such that one could evaluate whether the semantic information from the LR image is sufficient as a prior for HR reconstruction?**
> > In our method, *InterLCM*, we utilize a Visual Module to extract semantic information from LQ images for HQ reconstruction. As shown in **Figure.17** (top, the first row), we provide our model with LQ images exhibiting varying levels of degradation, decreasing from left to right. Notably, the semantic information from the LQ image suffices as a prior for HQ reconstruction when the degradation level is below a specific threshold (e.g., top, the third to fifth columns).
> >
> > To further assess the importance of facial semantic information from the LQ image for HQ reconstruction, we supplied the Visual Module with non-facial semantic images (e.g., a image featuring a tree or a solid color, shown in the bottom, first and second columns) and unrelated facial images (bottom, third and fourth columns). Using non-facial semantic images resulted in reconstructed outputs with white-point artifacts, whereas unrelated facial images provided sufficient semantic priors for generating HQ reconstructions with facial features.
> >
> > Furthermore, we observe that when the HQ image is used as the input to the Visual Module, the reconstructed image displays similar semantic information to that obtained using the LQ image (the fifth column). This finding indicates that the LQ image provides semantic information similar to that of the HQ image.
> >
> > **Q2: How were the hyperparameters (2) chosen in this study?**
> > Following CodeFormer[1], where they also apply the perceptual and adversarial losses, we set the $\lambda=0.8$ by default. This setup works the best for our BFR model in practice.
> >
> > [1] Towards Robust Blind Face Restoration with Codebook Lookup Transformer, NeurIPS 2022
> >
> > **Q3: Why was a four-step LCM considered instead of a different number of steps?**
> >  LCM employs a four-step inference process to balance image quality and inference time, as recommended by the original paper. In this paper, we use the recommended 4-step LCM model, while we also offer an ablation study with a 2-step LCM model. As observed from the table below, the 4-step LCM only works slightly worse than the 2-step LCM on two metrics, which is utilized as the backbone for our *InterLCM*.
> >
> >  |         | Celeba-Test        |          |          |          |                |          | LFW-Test       |          | WebPhoto-Test |          |WIDER-Test|          |
> > |------------------------------|--------------------|----------|----------|----------|----------------|----------|----------------|----------|---------------|----------|----------|----------|
> > |         | LPIPS↓             | FID↓     | MUSIQ↑   | IDS↓     | PSNR↑          | SSIM↑    | FID↓           | MUSIQ↑   | FID↓          | MUSIQ↑   | FID↓     | MUSIQ↑   |
> > |Ours (2step LCM)| 0.248 | 49.19 | 75.31| 34.92  | 23.91 | 0.662 | 56.21 |**76.24** | 75.84 | **76.11**| 38.23    | 76.00    |
> > |Ours (4step LCM)| **0.223**  | **45.38**| **76.58**| **33.64**| **25.19** | **0.718**| **51.32**  |76.16  | **75.48** | 75.88| **35.43**| **76.29**|

---

> ### Author Response · Authors · 2024-11-25
> **Response to Reviewer 43VF (3/3)**
>
> **Q4: Figure.1 is unclear. It says that LCM maps directly to the real image space. Does it imply LCM learns a mapping from LR to HR images directly like [2], rather than progressively denoising as depicted in Figures.2 and 3?**
>  LCM mechanism is different from [2]. The Consistency Model (CM) represents a distinct family of diffusion models. Unlike DDPM models, which perform denoising incrementally through multiple steps, CM directly maps $z_t$ to its original real image space $z_0$. As illustrated in **Figure.2**, the process begins by predicting the original image from random noise in the first step. In subsequent steps, additional noise is added to the previously produced image, and the model predicts the original image again from this noisy input.
>
> [2] I2SB: Image-to-Image Schrödinger Bridge.
>
> **Q5: What is the degradation process considered in the synthetic dataset? Is it a simple interpolation?**
> We detail our degradation process in **Appendix A.2**, which follows a typical procedure outlined in existing the state-of-the-art BFR work CodeFormer[A]:
>
> $x\_{l}=\[{[(x\_{h} * k\_{\sigma}){\downarrow\_{s}} + n\_{\delta}]\_{\text{JPEG}\_{q}}\]}{\uparrow\_{s}}$
>
> where $x_h$ and $x_l$ represent the HQ and LQ images, respectively, $k_\sigma$ is the Gaussian kernel with $\sigma\in\{1:15\}$, $\downarrow_s$ represents the downsampling operation with a scale factor $s\in\{1:30\}$, and $n_\delta$ denotes Gaussian noise with a standard deviation of $\delta\in\{0:20\}$. The convolution operation is denoted by $*$, followed by JPEG compression with a quality factor of $q\in\{30:90\}$. Finally, an upsampling operation $\uparrow_s$ with scale $s$ is applied to restore the original resolution of $512\times512$.
>
> [A] Towards robust blind face restoration with codebook lookup transformer, NeurIPS 2022

---

> > ### Comment · Reviewer_43VF · 2024-11-25
> >
> > Many thanks to the authors for their detailed response, which has addressed some of my concerns. However, I find that some of my concerns remain unresolved. In particular, my comments regarding the novelty of the work still stand. Therefore, I will maintain my original score. Please see below my comments.
> >
> > W1 & W6: My concern is that using better initialization from a different estimator is not a novel concept. The authors, in their response, argue that existing methods face issues with distribution gaps and highlight that LCM addresses this by mapping noise directly to real images. Firstly, in the multi-step case, LCM still alternates between denoising and noise injection steps. More importantly, I am unsure how this new argument adequately addresses my concern nor how LCM can address the distribution gaps. While I appreciate the point that extracting semantic and visual information from LR is a contribution, I struggle to see other significant contributions in this paper.
> >
> > W4: I acknowledge that the proposed method performs quantitatively better in simulations. However, in real cases—which are central to BFR—I do not observe a notable improvement in visual results. For instance, in Figure 7, particularly in the top two setups, it would be difficult to distinguish whether RestoreFormer, CodeFormer, or the proposed method performs better if the labels were blinded. Even when the ground truth is available, the proposed method appears to over-sharpen the results.
> >
> > W5: Regarding the response to the second statement I mentioned, predicting x0 or noise in diffusion models is effectively equivalent, making it possible to apply perceptual loss in any diffusion model. Therefore, it is not a convincing response. Please see “Three Equivalent Interpretations” in https://arxiv.org/pdf/2208.11970 for more details.
> >
> > Q1: The layout of the new Figure 17 appears somewhat unclear. I am unsure if the authors fully understood my question. My intent was to ask whether LR provides meaningful semantic information compared to its corresponding HR. Here, LR and HR are paired. However, the new figure seems to use unpaired HR for extracting semantic information, which does not address my question directly.
> >
> > Q4: The response is clear and aligns with my understanding of LCM. In this case, would it be more precise to phrase it as, “the network used in LCM maps …”?

---

> ### Author Response · Authors · 2024-11-28
> **New Response to Reviewer 43VF (1/6)**
>
> Thank you for your feedback and the opportunity to further clarify our work.
>
> **W1&W6(1)： My concern is that using better initialization from a different estimator is not a novel concept.**
>
> Using a better initialization is not the primary contribution of our method. More importantly, our approach leverages the inherent semantic consistency properties of LCM, as observed in our experiments. A good initialization alone is not enough to ensure good performance. We test SD Turbo as the backbone to develop our method for the BFR problem. In detail, we use a 4-step SD Turbo pretrained model, where the LQ image is treated as the input at an intermediate step of the SD Turbo process. The HQ image is then obtained by performing the remaining denoising steps (i.e., 3 steps) in the 4-step SD Turbo model. Unlike LCM, which incorporates a noise injection step during inference, SD Turbo only includes denoising steps and no noise addition step during inference. Therefore, we adopt the noise injection step from the diffusion training process. We evaluated the initial performance of LCM and SD Turbo on the COCO2014 and COCO2017 datasets. As shown in **Table A**, SD Turbo outperformed LCM in terms of FID and CLIPScore on both datasets as the initialization points. Compared to LCM, SD Turbo demonstrated superior generative capabilities. Therefore, SD Turbo should provide a better initialization.
>
>
> **Table.A The initial performance of SD Turbo and LCM on COCO2014 and COCO2017**
> | Dataset  | COCO2014-30K|          | COCO2017-5K|            |
> |----------|-------------|----------|------------|------------|
> | Metrics  | FID↓        |CLIPScore↑| FID↓       | CLIPScore↑ |
> | SD Turbo | **16.14**       |**0.33**      | **26.14**      | **0.33**   |
> | LCM      | 23.21       |0.26      | 40.37      | 0.30       |
>
> However, the quantitative analysis presented in **Table.B** indicates that SD Turbo is not optimally suited for BFR problems. For instance, SD Turbo fails to achieve state-of-the-art (SOTA) performance across all metrics; it only attains the second-best score on the MUSIQ metric for the real-world WebPhoto-Test dataset. In contrast, our approach, which incorporates LCM, surpasses SD Turbo in all metrics on both synthetic and real-world datasets, with the exception of the MUSIQ metric on the WebPhoto-Test dataset.

---

> ### Author Response · Authors · 2024-11-28
> **New Response to Reviewer 43VF (2/6)**
>
> **Table.B Quantitative comparison on the synthetic and real-world dataset**
> | Dataset   |Synthetic dataset|  |   |  |    |    |Real-world datasets |   |   |  |    |   |
> |-----------|-----------------|--|---|--|----|----|---|--|----|---|----|---|
> | Metrics   | Celeba-Test     |  |   |  |    |    | LFW-Test |  | WebPhoto-Test|  | WIDER-Test|    |
> | Method    | LPIPS↓          | FID↓    | MUSIQ↑     | IDS↑       | PSNR↑      | SSIM↑      | FID↓     |  MUSIQ↑  |   FID↓  |  MUSIQ↑  |   FID↓  | MUSIQ↑  |
> | Input     | 0.574           | 145.22  | 72.81      | 47.94      | 22.72      | 0.706      | 138.87      | 26.87      | 171.63       | 18.63 | 201.31    | 14.22  |
> | PULSE     | 0.356           | 68.33   | 66.46      | 43.98      | 22.10       | 0.592     | 67.01       | 65.00      | 85.69        | 63.88 | 70.65     | 63.01  |
> | DFDNet    | 0.332           | 54.21   | 72.08      | 40.44      | 24.27       | 0.628     | 60.28       | 73.06      | 92.71        | 68.50 | 59.56     | 62.02  |
> | PSFRGAN                | 0.294             | 54.21   | 73.32      | 39.63      | 24.66       | 0.661     | 49.89       | 73.60      | 85.42        | 71.67 | 54.52     | 71.50  |
> | GFPGAN                 | 0.230             | 49.84   | 73.90      | _34.56_    | 24.64       | 0.688     | 50.36       | 73.57      | 87.47        | 72.08 | 39.45     | 72.79  |
> | GPEN                   | 0.290             | 63.44   | 67.52      | 36.17      | **25.48**   | _0.708_   | 61.04       | 68.96      | 99.09        | 61.10 | 46.25     | 62.64  |
> | RestoreFormer          | 0.241             | 50.04   | 73.85      | 36.16      | 24.61       | 0.660     | 48.77       | 73.70      | 80.85        | 69.83 | 50.04     | 67.83  |
> | VQFIR                  | 0.245             | _41.84_ | 75.18      | 35.74      | 24.06       | 0.666     | 51.33       | 71.74      | _75.77_      | 72.02 | 44.09     | _74.01_  |
> | CodeFormer             | _0.227_           | 52.94   | _75.55_    | 37.27      | 25.15       | 0.685     | 52.84       | _75.48_    | 83.95        | 74.00 | 39.22     | 73.41  |
> | DR2                    | 0.264             | 54.48   | 67.99      | 44.00      | 25.03       | 0.617     | _45.71_     | 71.50      | 109.24       | 62.37 | 48.20     | 60.28  |
> | DiffFace               | 0.272             |**39.23**| 68.87      | 45.80      | 24.80       | 0.684     | 46.31       | 69.76      | 80.86        | 65.37 | 37.74     | 65.02  |
> | PGDiff                 | 0.300             | 47.26   | 71.81      | 55.90      | 22.72       | 0.659     | **44.65**   | 71.74      | 101.68       | 67.92 | 38.38     | 68.26  |
> | WaveFace               | –                 | –       | –          | –          | –           | –         | 53.88       | 73.54      | 78.01        | 70.45 | _37.23_   | 68.29  |
> | *Ours (SD Turbo)*        | 0.257             | 48.51   | 74.15      | 37.02      | 23.30       | 0.666     | 56.44       | 74.24      | 84.66        |_74.41_| 43.53     | 72.35  |
> | *Ours (LCM)*             | **0.223**         | 45.38   | **76.58**  | **33.64**  | _25.19_     | **0.718** | 51.32       | **76.16**  | **75.48**    |**75.88**|**35.43**|**76.29**|

---

> ### Author Response · Authors · 2024-11-28
> **New Response to Reviewer 43VF (3/6)**
>
> **W1&W6(2)： The authors, in their response, argue that existing methods face issues with distribution gaps and highlight that LCM addresses this by mapping noise directly to real images. Firstly, in the multi-step case, LCM still alternates between denoising and noise injection steps. More importantly, I am unsure how this new argument adequately addresses my concern nor how LCM can address the distribution gaps. While I appreciate the point that extracting semantic and visual information from LR is a contribution, I struggle to see other significant contributions in this paper.**
>
> **Noise injection**: During the training phase of LCM, noise is injected into the real image, and then directly maps any point in the ODE trajectory back to its origin, facilitating semantic consistency generation compared to SD, as also shown in Figure.1. In the inference phase of LCM, the same noise injection step used during training is inherited, while the inference of SD does not use a noise injection like in its training phase. Existing work [1,2] uses noise injection that the inference of SD does not include. [A] find that a bias exists between the latent distribution in the vanilla generation chain and the image noise injection chain, resulting in a significant domain gap between the denoised images. In contrast, the LCM inference already includes a noise injection step (see **Figure.2.**).
>
> We utilize the semantic consistency properties (**Figure.1**) and the noise injection step during inference (**Figure.2**). Additionally, thanks to the few-step generation of LCM, we can integrate the perception loss in the image space while avoiding the distribution gap when injecting noise into the LQ image. It is more suitable to combine with perception loss. Additionally, to the best of our knowledge, we are the first to leverage the consistency properties (see **Figure.1**) of LCM for exploring BFR tasks.
>
> From the philosophical aspect of machine learning, a cornerstone principle is to train algorithms under the same circumstances as they are tested. We therefore think, that the fact that we can actually train our network with the LQ image (which is applied during inference) is an advantage, and could explain the excellent results we report in our experiments.
>
> [A] Real-World Image Variation by Aligning Diffusion Inversion Chain. \
> [1] Come-closer-diffuse-faster: Accelerating conditional diffusion models for inverse problems through stochastic contraction. \
> [2] I2SB: Image-to-Image Schrödinger Bridge.

---

> ### Author Response · Authors · 2024-11-28
> **New Response to Reviewer 43VF (4/6)**
>
> **W4: I acknowledge that the proposed method performs quantitatively better in simulations. However, in real cases—which are central to BFR—I do not observe a notable improvement in visual results. For instance, in Figure 7, particularly in the top two setups, it would be difficult to distinguish whether RestoreFormer, CodeFormer, or the proposed method performs better if the labels were blinded. Even when the ground truth is available, the proposed method appears to over-sharpen the results.**
> Thanks for your acknowledge that the proposed method performs quantitatively better in simulations.
>
> We also want to emphasize that the blind face restoration is based on the whole dataset statistics, not merely focusing on improving the restoration quality of a single person. As shown in **Table.1**, for the evaluation on real-world datasets, our method achieves the best performance for three datasets on two metrics: FID and MUSIQ, except for the FID value on the LFW-Test dataset.
>
> **Qualitative results**: In **Figure.7**, to show that our method significantly outperforms baselines in reconstructed textual details, we zoomed in on other areas of detail.  For example, in the first case (the first and second rows), our method can reconstruct the better detail in hair (red box), and eyes with the surrounding details (green box), while the baselines (including RestoreFormer and CodeFormer) generate blurr hair and meet artifacts for eyes and its surrounding details. In the second case (the third and fourth rows), our method can reconstruct more realistic crow's feet (red box) and hair details (green box). In contrast, the baseline methods, including RestoreFormer and CodeFormer, generate blurry crow's feet and hair. Although DifFace is capable of generating detailed hair, it exhibits noticeable artifacts around the eyes and in the wrinkles.
> We present additional qualitative comparisons of the baselines on real-world images from the LFW-Test, WebPhoto-Test, and WIDER-Test datasets in **Figure.23**. As shown in **Figure.23**, our method can reconstruct more realistic details in forehead wrinkles (first and second rows), eyes and eyebrows (third and fourth rows), and hair (fifth and sixth rows). These results demonstrate that our method outperforms the baselines in real-world scenarios.
>
> **Quantitative results**: Furthermore, we provide the top two setups with the **MUSIQ** metric of **Figure.7**. Our method achieves the best MUSIQ scores in both setups compared to the baselines, indicating that our results have the highest perceptual quality.
> | Method       | Setup 1 | Setup 2 |
> |--------------|---------|---------|
> | Input        | 19.06   | 13.42   |
> | PULSE        | 62.69   | 73.13   |
> | DFDNet       | 69.32   | 76.02   |
> | PSFRGAN      | 72.31   | 76.80   |
> | GFPGAN       | 69.37   | 77.79   |
> | GPEN         | 66.68   | 66.53   |
> | RestoreFormer| 70.45   | 73.57   |
> | VQFR         | 69.76   | 75.50   |
> | CodeFormer   | 72.39   | 76.48   |
> | DR2          | 69.66   | 63.46   |
> | DifFace      | 60.22   | 73.88   |
> | PGDiff       | 69.84   | 76.32   |
> | WaveFace     | 65.17   | 72.76   |
> | **Ours**     | **72.83** | **78.05** |

---

> ### Author Response · Authors · 2024-11-28
> **New Response to Reviewer 43VF (5/6)**
>
> **W5: Regarding the response to the second statement I mentioned, predicting x0 or noise in diffusion models is effectively equivalent, making it possible to apply perceptual loss in any diffusion model. Therefore, it is not a convincing response. Please see “Three Equivalent Interpretations” in https://arxiv.org/pdf/2208.11970 for more details.**
>
> **Firstly**, we agree with your point that predicting $x_0$ or noise is theoretically equivalent. However, in practice, the distribution gap [D] (e.g., **Figure.2** in [D]) and signal leakage [E] (e.g., **Figure.1** in [E]) accumulate because the training images introduce image priors during the training process. As a result, $x_T$ does not strictly follow a normal Gaussian distribution. This discrepancy underscores why the consistency model focuses on directly mapping $x_t$ to $x_0$, effectively bridging the gap and mitigating these issues.
>
> [D] Real-world image variation by aligning diffusion inversion chain. NeurIPS2023 \
> [E] Exploiting the signal-leak bias in diffusion models. WACV2024
>
> **Secondly**, we want to claim that LCM was originally proposed to speed-up DM computation. However, in this paper, we show that there is another aspect of LCM which is very crucial for BFR problem: it allows for true end-to-end training for image generation, whereas DMs only optimize single denoising steps. The end-to-end training is also shown in Figure 3 where we can apply losses after the n-step generation of the final image, and backpropagate through all the image generation steps. This end-to-end nature of LCM allows us to do things which are not possible with standard DMs (where only singel step from $x_t$ to $x_{t-1}$ is optimized).
>
> As a practical usage in this paper, it allows us to train with **perceptual losses** in the image domain, which is crucial for BFR where faithfulness is of utmost importance (compared to general image generation tasks). We can apply the perceptual and adversarial losses directly in the image domain and backpropagate through all image generation steps. It is unclear how to introduce a perceptual loss in a traditional DM training, where the noise-estimation of a single time step is optimized at the time. In this case, we have no results in the image domain on which to apply the perceptual loss. Another alternative ,as you mentioned, could be to estimate $x_0$ also from the intermediate steps, however, this would yield perceptually very low-quality images for most time steps. In this case, the perceptual loss is not expected to work due to the low semantic consistency and worse image quality. In this case, we would like to claim that predicting $x_0$ is not perpectly equivalent to predicting the noise.
>
> As a conclusion, the usage of LCM allows training with losses directly in the image domain; to the best of our knowledge, we are the first to show this.

---

> ### Author Response · Authors · 2024-11-28
> **New Response to Reviewer 43VF (6/6)**
>
> **Q1: The layout of the new Figure 17 appears somewhat unclear. I am unsure if the authors fully understood my question. My intent was to ask whether LR provides meaningful semantic information compared to its corresponding HR. Here, LR and HR are paired. However, the new figure seems to use unpaired HR for extracting semantic information, which does not address my question directly.**
>
> We update **Figure.19** with paired HQ and HQ images input for our method of the updated submission.
> In our method (**Figure.19**, top(a)), InterLCM, we utilize a Visual Module to extract semantic information from LQ images for HQ reconstruction. To demonstrate that the LQ image suffices to provide a prior for HQ reconstruction, we provide our model with LQ images exhibiting varying levels of degradation, decreasing from left to right (**Figure.19**, middle(the first row)). The reconstruction results (**Figure.19**, middle(the second row)) show that the semantic information from the LQ image suffices as a prior for HQ reconstruction when the degradation level is below a specific threshold (**Figure.19**, middle(the third to fifth columns)).
>
>
> Meanwhile, we observe that when the HQ image is used as the input to both the Visual Module and Spatial Encoder (**Figure.19**, top(b)), the reconstructed image displays similar semantic information to that obtained using the LQ image (**Figure.19**, bottom(the first column)). This result further indicates that the LQ image provides semantic information similar to that of the HQ image (**Figure.19**, middle(the last column) vs., bottom(the first column)).
>
>
> Then, we verify the provision of paired LQ and HQ images, which are provided to the Visual Module and Spatial Encoder (**Figure.19**(c)). We also obverse that the reconstructed result shows similar semantic information to the HQ image (**Figure.19**, bottom(the second column)).
>
>
> To further assess the importance of facial semantic information from the LQ image for HQ reconstruction, we supplied the Visual Module with non-facial semantic images (**Figure.19**, top(d)), such as non-facial semantic images (e.g., a image featuring a tree or a solid color) and unrelated facial images (**Figure.19**, bottom(third and fifth columns)). Using non-facial semantic images resulted in reconstructed outputs with artifacts (**Figure.19**, bottom(third and fourth columns)), whereas unrelated facial images provided sufficient semantic priors for generating HQ reconstructions with facial features (**Figure.19**, bottom(fifth columns)).
>
>
> **Q4: The response is clear and aligns with my understanding of LCM. In this case, would it be more precise to phrase it as, “the network used in LCM maps …”?**
> Yes, we are pleased that we have reached alignment on this point. Based on your suggestion, we have already updated the caption of **Figure.1** accordingly.

---

> ### Author Response · Authors · 2024-12-02
>
> Dear Reviewer **43VF**,
>
> Thank you once again for your constructive comments. Your constructive comments have been very helpful.
>
> In response to your remaining questions, we have conducted experiments to clarify that **(W1&W6(1))** using a better initialization (i.e., SD Turbo) is not optimally suited for BFR problems and is not the primary contribution of our method. We have also clarified ****(W1&W6(2))**** the distribution gaps, **(W5)** the prediction of $x_0$, and **(Q1)** provided additional explanations for **Figure 17**. **(W4)** Additionally, we zoomed in on other areas of detail in **Figure.7** and provide the additional visual results for real cases.  We would like to know if our responses have addressed your concerns.
>
> **As the rebuttal window is closing soon, we genuinely appreciate your feedback. We hope our response has addressed your concerns.**
>
> Best Regards,
> Authors of submission 536

---

### Official Review · Reviewer_FpYB · 2024-11-05

**Soundness:** 3
**Presentation:** 4
**Contribution:** 3
**Rating:** 6
**Confidence:** 4

**Summary:**

The authors propose using the LQ image as an intermediate state in the LCM. Because of how LCM works, they can apply image-based loss during training. The method uses the diffusion model’s prior by adding noise to the LQ image. This drags the LQ image to the domain of generative capability.  And then they guide diffusion model to restore the LQ image with constraint of semantic related info.

**Strengths:**

(+) The proposed method is carefully designed and based on an interesting observation about LCM.

(+) The paper is well-written and easy to follow. The figures are well-designed and professionally presented, clearly illustrating the main takeaway.

(+) The experiments are thorough and solid, with especially useful discussions in the appendix.

(+) The performance is solid, outperforming baselines in terms of visual quality on real-world images across many cases.

(+) Thanks to the advantages of LCM, the inference speed is excellent, which could benefit the related community.

**Weaknesses:**

(-) What does the author mean by "back to the low-quality image initialization" in line 100? Does Figure 3 show all the trainable components? It seems the authors are suggesting that some latents are also trainable.

(-) Table 2 should be reorganized. The current version is squeezed and difficult to read.

(-) It seems the whole pipeline has a strong capability to preserve identity. I am curious about how the method performs when the face in the LQ image has additional textures, like tattoos or festival-style face paint. How would the proposed method handle these cases?

(-) Does the author's method design draw significant inspiration from general image restoration approaches built on diffusion models? I ask this because using a ControlNet-like design to extract useful information from low-quality images is not a new concept. Meanwhile, it seems that the main performance improvement comes from the spatial encoder, which is the ControlNet as semantic info encoder, as demonstrated in Tables 1 and 2. Additionally, the second main source of performance appears to be the LPIPS loss. It seems that one could solely rely on the spatial encoder to achieve good results. Could I understand that the performance improvements on your task could be easily obtained by simply using a ControlNet to extract LQ's semantic info? If this is not the case, please provide a justification.

**Questions:**

All my concerns are listed in the weakness part. The main concern is about the key source of performance, which requires the author’s justification for the method design. I will consider raising my score if the author provides a thorough explanation.

---

> ### Author Response · Authors · 2024-11-25
> **Response to Reviewer FpYB (1/1)**
>
> We appreciate your constructive comments and will incorporate the discussions mentioned below to enhance the quality of our paper.
>
> **W1: What does "back to the low-quality image initialization" in line 100 mean? Does Figure 3 show all trainable components?**
> In our method *InterLCM*, by integrating both perceptual loss and adversarial loss, backpropagation is performed from the real image $z_0$ to the initial low-quality image $x_{\tau_n}$. During this process, the Visual Encoder (VE) and Spatial Encoder (SE) are updated using the computed gradients. In contrast, the diffusion loss commonly used in ControlNet and standard diffusion model training performs backpropagation from $z_{t-1}$​ to $z_t$, with the primary goal of updating the Diffusion-UNet. Notably, in our approach, the latent variables remain unchanged throughout the process.
>
> We already rewrote the corresponding sentence in Line 100 for clarity.
>
> **W2: Table.2 should be reorganized. The current version is squeezed and difficult to read.**
> Thanks to your advice. We reorganize and enlarge the font of the table in the updated version.
>
> **W3: The method's performance on LQ faces with textures like tattoos or face paint.**
> As shown in **Figure.16** (the third and fourth rows), our method, *InterLCM*, demonstrates the ability to reconstruct high-quality details even in challenging cases, such as images featuring tattoos or festival-style face paint. However, when tattoos contain intricate details, such as text (e.g., the last column), accurately recovering these ambiguous elements during high-quality face reconstruction becomes challenging. This limitation may stem from the scarcity of such textures in the training dataset.
> An illustration of the complex textures in our training dataset FFHQ is also shown in **Figure.16** (the first and second rows), where the festival-style face paints and rich-color hair appear multiple times during training.
>
> **W4(1): Does the author's method design draw significant inspiration from general image restoration approaches built on diffusion models (e.g., ControlNet) ?**
> This work is inspired by the observation that latent consistency models (LCMs) demonstrate greater semantic consistency in subject identity, spatial structure, and color preservation. However, general diffusion models often exhibit inferior semantic consistency, including issues with identity preservation, structural stability, and color consistency, as demonstrated in **Figure.1**.
>
> Additionally, *InterLCM* enables the integration of image-level losses, such as perceptual loss and adversarial loss. In contrast, using ControlNet with the original diffusion loss (Eq. 1) proves ineffective for face restoration, as evidenced in **Table 4**. Instead, leveraging the ControlNet architecture—replicated from the Stable Diffusion encoder modules and serving as the Spatial Encoder (SE) in our method, *InterLCM*—alongside the perceptual loss (Eq. 2) significantly improves blind face restoration (BFR) performance.
>
> **W4(2): It seems that the main performance improvement comes from the spatial encoder.**
> We employ a Visual Encoder to extract semantic information from faces. Initially, faces are embedded by the CLIP image encoder as image embeddings, providing the Latent Consistency Model (LCM) with face-specific semantic priors. To preserve structural integrity, we utilize a Spatial Encoder. As illustrated in Fig. 8③, using only the Spatial Encoder can effectively maintain the structural information of the face but fails to reconstruct finer facial details, such as the eyes. These intricate details primarily rely on the Visual Encoder, as evidenced by the results when solely using the Visual Encoder (Fig. 8①).
>
> **W4(3): The second main source of performance appears to be the LPIPS loss.**
>  As mentioned above, our approach does not utilize the LPIPS loss. Instead, we are the first to integrate perceptual loss and adversarial loss into diffusion-based blind face restoration (BFR) solutions. The effectiveness of this integration is demonstrated in the ablation studies provided in **Figure.9** and **Table.3**.
>
>  **W4(4): Could the performance improvements be easily achieved by using a ControlNet alone?**
>  Naively applying ControlNet for blind face restoration, as shown in **Figure.9** (the fifth to seventh columns), uses the low-quality (LQ) image as input. While it can generate high-quality outputs that preserve structural integrity, it often compromises fidelity.

---

> > ### Comment · Reviewer_FpYB · 2024-11-26
> >
> > Thanks for authors response, which addressed most of my concerns and questions. I have also carefully reviewed the comments from the other reviewers.
> >
> > Regarding the qualitative and quantitative results, I acknowledge the improvements presented. For instance, in figure 7 (top 2), the improvement is clearly noticeable. However, the novelty of the work remains some questions. I am curious how the authors address Weaknesses 1 and 6 mentioned in Reviewer 43VF's updated comments

---

> ### Author Response · Authors · 2024-11-28
> **New Response to Reviewer FpYB (1/3)**
>
> Thank you for acknowledging our quantitative and qualitative evaluation results. This acknowledgement is very important to us. For your convenience, we have provided below the weaknesses 1 and 6 from Reviewer 43VF, along with our responses.
>
> **W1&W6(1) (Reviewer 43VF)： My concern is that using better initialization from a different estimator is not a novel concept.**
>
> Using a better initialization is not the primary contribution of our method. More importantly, our approach leverages the inherent semantic consistency properties of LCM, as observed in our experiments. A good initialization alone is not enough to ensure good performance. We test SD Turbo as the backbone to develop our method for the BFR problem. In detail, we use a 4-step SD Turbo pretrained model, where the LQ image is treated as the input at an intermediate step of the SD Turbo process. The HQ image is then obtained by performing the remaining denoising steps (i.e., 3 steps) in the 4-step SD Turbo model. Unlike LCM, which incorporates a noise injection step during inference, SD Turbo only includes denoising steps and no noise addition step during inference. Therefore, we adopt the noise injection step from the diffusion training process. We evaluated the initial performance of LCM and SD Turbo on the COCO2014 and COCO2017 datasets. As shown in **Table A**, SD Turbo outperformed LCM in terms of FID and CLIPScore on both datasets as the initialization points. Compared to LCM, SD Turbo demonstrated superior generative capabilities. Therefore, SD Turbo should provide a better initialization.
>
>
> **Table.A The initial performance of SD Turbo and LCM on COCO2014 and COCO2017**
> | Dataset  | COCO2014-30K|          | COCO2017-5K|            |
> |----------|-------------|----------|------------|------------|
> | Metrics  | FID↓        |CLIPScore↑| FID↓       | CLIPScore↑ |
> | SD Turbo | **16.14**       |**0.33**      | **26.14**      | **0.33**   |
> | LCM      | 23.21       |0.26      | 40.37      | 0.30       |
>
> However, the quantitative analysis presented in **Table.B** indicates that SD Turbo is not optimally suited for BFR problems. For instance, SD Turbo fails to achieve state-of-the-art (SOTA) performance across all metrics; it only attains the second-best score on the MUSIQ metric for the real-world WebPhoto-Test dataset. In contrast, our approach, which incorporates LCM, surpasses SD Turbo in all metrics on both synthetic and real-world datasets, with the exception of the MUSIQ metric on the WebPhoto-Test dataset.

---

> ### Author Response · Authors · 2024-11-28
> **New Response to Reviewer FpYB (2/3)**
>
> **Table.B Quantitative comparison on the synthetic and real-world dataset**
> | Dataset   |Synthetic dataset|  |   |  |    |    |Real-world datasets |   |   |  |    |   |
> |-----------|-----------------|--|---|--|----|----|---|--|----|---|----|---|
> | Metrics   | Celeba-Test     |  |   |  |    |    | LFW-Test |  | WebPhoto-Test|  | WIDER-Test|    |
> | Method    | LPIPS↓          | FID↓    | MUSIQ↑     | IDS↑       | PSNR↑      | SSIM↑      | FID↓     |  MUSIQ↑  |   FID↓  |  MUSIQ↑  |   FID↓  | MUSIQ↑  |
> | Input     | 0.574           | 145.22  | 72.81      | 47.94      | 22.72      | 0.706      | 138.87      | 26.87      | 171.63       | 18.63 | 201.31    | 14.22  |
> | PULSE     | 0.356           | 68.33   | 66.46      | 43.98      | 22.10       | 0.592     | 67.01       | 65.00      | 85.69        | 63.88 | 70.65     | 63.01  |
> | DFDNet    | 0.332           | 54.21   | 72.08      | 40.44      | 24.27       | 0.628     | 60.28       | 73.06      | 92.71        | 68.50 | 59.56     | 62.02  |
> | PSFRGAN                | 0.294             | 54.21   | 73.32      | 39.63      | 24.66       | 0.661     | 49.89       | 73.60      | 85.42        | 71.67 | 54.52     | 71.50  |
> | GFPGAN                 | 0.230             | 49.84   | 73.90      | _34.56_    | 24.64       | 0.688     | 50.36       | 73.57      | 87.47        | 72.08 | 39.45     | 72.79  |
> | GPEN                   | 0.290             | 63.44   | 67.52      | 36.17      | **25.48**   | _0.708_   | 61.04       | 68.96      | 99.09        | 61.10 | 46.25     | 62.64  |
> | RestoreFormer          | 0.241             | 50.04   | 73.85      | 36.16      | 24.61       | 0.660     | 48.77       | 73.70      | 80.85        | 69.83 | 50.04     | 67.83  |
> | VQFIR                  | 0.245             | _41.84_ | 75.18      | 35.74      | 24.06       | 0.666     | 51.33       | 71.74      | _75.77_      | 72.02 | 44.09     | _74.01_  |
> | CodeFormer             | _0.227_           | 52.94   | _75.55_    | 37.27      | 25.15       | 0.685     | 52.84       | _75.48_    | 83.95        | 74.00 | 39.22     | 73.41  |
> | DR2                    | 0.264             | 54.48   | 67.99      | 44.00      | 25.03       | 0.617     | _45.71_     | 71.50      | 109.24       | 62.37 | 48.20     | 60.28  |
> | DiffFace               | 0.272             |**39.23**| 68.87      | 45.80      | 24.80       | 0.684     | 46.31       | 69.76      | 80.86        | 65.37 | 37.74     | 65.02  |
> | PGDiff                 | 0.300             | 47.26   | 71.81      | 55.90      | 22.72       | 0.659     | **44.65**   | 71.74      | 101.68       | 67.92 | 38.38     | 68.26  |
> | WaveFace               | –                 | –       | –          | –          | –           | –         | 53.88       | 73.54      | 78.01        | 70.45 | _37.23_   | 68.29  |
> | *Ours (SD Turbo)*        | 0.257             | 48.51   | 74.15      | 37.02      | 23.30       | 0.666     | 56.44       | 74.24      | 84.66        |_74.41_| 43.53     | 72.35  |
> | *Ours (LCM)*             | **0.223**         | 45.38   | **76.58**  | **33.64**  | _25.19_     | **0.718** | 51.32       | **76.16**  | **75.48**    |**75.88**|**35.43**|**76.29**|

---

> ### Author Response · Authors · 2024-11-28
> **New Response to Reviewer FpYB (3/3)**
>
> **W1&W6(2) (Reviewer 43VF)： The authors, in their response, argue that existing methods face issues with distribution gaps and highlight that LCM addresses this by mapping noise directly to real images. Firstly, in the multi-step case, LCM still alternates between denoising and noise injection steps. More importantly, I am unsure how this new argument adequately addresses my concern nor how LCM can address the distribution gaps. While I appreciate the point that extracting semantic and visual information from LR is a contribution, I struggle to see other significant contributions in this paper.**
>
> **Noise injection**: During the training phase of LCM, noise is injected into the real image, and then directly maps any point in the ODE trajectory back to its origin, facilitating semantic consistency generation compared to SD, as also shown in Figure.1. In the inference phase of LCM, the same noise injection step used during training is inherited, while the inference of SD does not use a noise injection like in its training phase. Existing work [1,2] uses noise injection that the inference of SD does not include. [A] find that a bias exists between the latent distribution in the vanilla generation chain and the image noise injection chain, resulting in a significant domain gap between the denoised images. In contrast, the LCM inference already includes a noise injection step (see **Figure.2.**).
>
> We utilize the semantic consistency properties (**Figure.1**) and the noise injection step during inference (**Figure.2**). Additionally, thanks to the few-step generation of LCM, we can integrate the perception loss in the image space while avoiding the distribution gap when injecting noise into the LQ image. It is more suitable to combine with perception loss. Additionally, to the best of our knowledge, we are the first to leverage the consistency properties (see **Figure.1**) of LCM for exploring BFR tasks.
>
> From the philosophical aspect of machine learning, a cornerstone principle is to train algorithms under the same circumstances as they are tested. We therefore think, that the fact that we can actually train our network with the LQ image (which is applied during inference) is an advantage, and could explain the excellent results we report in our experiments.
>
> [A] Real-World Image Variation by Aligning Diffusion Inversion Chain. \
> [1] Come-closer-diffuse-faster: Accelerating conditional diffusion models for inverse problems through stochastic contraction. \
> [2] I2SB: Image-to-Image Schrödinger Bridge.

---

> ### Author Response · Authors · 2024-12-02
>
> Dear Reviewer **FpYB**,
>
> Thank you once again for your constructive comments. Your suggestions regarding the clarification of the sentence (line 100), Table 2, and the method design have been very helpful. We would like to know if our responses have adequately addressed your concerns.
> We are also eager to hear any additional concerns or suggestions from you that could help us further strengthen our paper. Your feedback would be invaluable in making this work even more compelling and worthy of acceptance.
>
> **As the rebuttal window is closing soon, we genuinely appreciate your feedback. We hope our response has addressed your concerns.**
>
> Best Regards,
> Authors of submission 536

---

### Author Response · Authors · 2024-11-25
**General response to reviewers**

We appreciate all reviewers (**R1=FpYB**, **R2=43VF**, **R3=Hm1E**, **R4=6W3D**, **R5=aK8y**) for their positive feedback. They note that this paper is well-written (**R4**) and readability (**R3**); that the observation about LCM enjoying more semantic consistency (**R2**, **R3**) is interesting (**R1**); that the proposed method is a effective BFR method (**R3**) and framework (**R5**), maintains more semantic consistency (**R3**, **R5**); that the method outperform the baselines (**R1**, **R2**, **R3**); that experiments are thorough, solid (**R1**), comprehensive, well-organized (**R2**), and systematic (**R4**), as well as reasonable and sufficient (**R3**); that the inference is faster than stable diffusion models (**R1**, **R5**).

**We sincerely thank all the reviewers for their thoughtful and constructive feedback.
The revised part in the resubmission paper is highlighted in blue for your reference.** Below we respond to questions raised by reviewers.

We use **W** to abbreviate Weaknesses and **Q** to represent Questions.

---

### Meta-Review · Area_Chair_KKix · 2024-12-16

**Metareview:**

This paper introduces InterLCM, a novel approach to blind face restoration (BFR) that leverages the strengths of Latent Consistency Models (LCMs). The authors argue that LCMs, with their superior semantic consistency and efficiency, address the limitations of traditional diffusion models in BFR tasks. The core of InterLCM lies in treating low-quality images as an intermediate state in the LCM process, enabling the integration of perceptual loss during training. The method also incorporates visual and spatial modules to enhance the fidelity of the restored images.

Strengths:
- The paper is well-written and easy to follow.
- The proposed method is based on an interesting observation about LCMs.
- The experiments are thorough and solid, with especially useful discussions in the appendix.
- The performance is solid, outperforming baselines on real-world images across many cases.
- The inference speed is very good, thanks to the advantages of LCMs.

Weaknesses:
- The novelty of the work is questioned, as it combines many existing techniques.
- The performance improvements seem to come mainly from the spatial encoder and the LPIPS loss, raising questions about the necessity of other components.

The paper presents a well-motivated and technically sound approach to BFR, and the results are clearly good. While the novelty is not groundbreaking, the improvements are evident, and the robustness on challenging cases is valuable.

**Additional Comments On Reviewer Discussion:**

During the rebuttal period, the reviewers raised several important points:

- Novelty: Reviewer 43VF and 6W3D questioned the novelty of the approach, pointing out similarities to existing techniques. The authors clarified that their main contribution lies in the specific way they leverage LCMs and integrate perceptual loss for BFR.
- Performance improvements: Reviewer FpYB asked about the key source of performance improvement. The authors explained that while the spatial encoder and LPIPS loss contribute, the overall performance is a result of the combined approach.
- Generalization: Reviewers Hm1E, 6W3D, and aK8y raised concerns about the method's limited scope, focusing only on face restoration. The authors acknowledged this limitation and suggested potential extensions to broader image restoration tasks.

For the final decision, I weighed the strong quantitative and qualitative results more heavily than the concerns about novelty. The authors' clarifications and additional experiments during the rebuttal period also helped address some of the reviewers' concerns.

---

### Decision · Program_Chairs · 2025-01-22

Accept (Poster)